# Enhancing LLM's Cognition via Structurization

**Kai Liu**[1,2*], **Zhihang Fu**[2†], **Chao Chen**[2], **Wei Zhang**[1], **Rongxin Jiang**[1†],
**Fan Zhou**[1], **Yaowu Chen**[1], **Yue Wu**[2], **Jieping Ye**[2]

[1]Zhejiang University,  [2]Alibaba Cloud

## Abstract

When reading long-form text, human cognition is complex and structurized. While large language models (LLMs) process input contexts through a causal and sequential perspective, this approach can potentially limit their ability to handle intricate and complex inputs effectively. To enhance LLM's cognition capability, this paper presents a novel concept of context *structurization*. Specifically, we transform the plain, unordered contextual sentences into well-ordered and hierarchically structurized elements. By doing so, LLMs can better grasp intricate and extended contexts through precise attention and information-seeking along the organized structures. Extensive evaluations are conducted across various model architectures and sizes (including a series of auto-regressive LLMs as well as BERT-like masking models) on a diverse set of NLP tasks (*e.g.*, context-based question-answering, exhaustive hallucination evaluation, and passage-level dense retrieval). Empirical results show consistent and significant performance gains afforded by a single-round structurization. In particular, we boost the open-sourced LLaMA2-70B model to achieve comparable performance against GPT-3.5-Turbo as the hallucination evaluator. Besides, we show the feasibility of distilling advanced LLMs' language processing abilities to a smaller yet effective StruXGPT-7B to execute structurization, addressing the practicality of our approach. Code is available at `https://github.com/alibaba/struxgpt`.

## 1 Introduction

Large language models (LLMs) have emerged with remarkable language capabilities [6, 53, 1], yet remain at a discernible distance from human-level intelligence, especially when handling long-form, sophisticated contexts as inputs [3, 36]. Scaling up the model size has significant benefits for boosting context-comprehension and instruction-following abilities for LLMs [47, 58]. However, it is generally resource-intensive on both model training and inference. This paper presents another perspective on enhancing LLMs' cognition capability without altering the models: context *structurization*.

The idea of structurization is motivated by neurocognitive science [51, 5, 17]. In human cognition, as indicated in Fig. 1, sophisticated text sequences will be processed and consolidated into a structured knowledge tree, with factual elements well-organized hierarchically [28, 15]. This process is defined as *structurization*. People can precisely search information from general concepts to specific details and make connections and comparisons along structures. We thus aim to transform plain texts into structurized inputs, helping LLMs recognize and understand contexts in a human manner [71].

As Fig. 1 illustrates, input sequences are reorganized in a simple but generic three-layer structure: *scope*, *aspects*, and *descriptions*. The scope summarizes the topic and contents, unfolding into several main aspects with corresponding detailed descriptions. The structurized results can be freely

---

*Work done during Kai Liu's research internship at Alibaba Cloud. Email: kail@zju.edu.cn.

†Corresponding authors. Email: rongxinj@zju.edu.cn, zhihang.fzh@alibaba-inc.com.

assembled into various natural language forms, depending on the specific downstream tasks. Fig. 2 provides an exemplar of our overall framework: after structurizing and reassembling the vanilla context to highlight its scope and aspects of data imbalance handling techniques, LLMs become able to grasp the target information about the dynamic weighting strategy and generate reliable responses.

We first execute the structurization by prompting advanced commercial LLMs (*e.g.*, GPT-3.5-Turbo[1] or Qwen-Max[2]) with a few examples, and then collect the results to train a smaller 7B-parameter LLaMA2 [53] or Qwen [2] model as our StruXGPT. This is motivated by [48], where the fundamental syntactic processing ability from giant LLMs is distilled into a responsive and affordable StruXGPT-7B. Comprehensive evaluations indicate that StruXGPT-7B inherits 97% of the structurization ability from the teacher model, showing our method's feasibility.

Empirical experiments are conducted on a diverse set of NLP tasks (*e.g.*, context-based question-answering, exhaustive hallucination evaluation, and passage-level dense retrieval). The results show that with a single-turn structurization by our StruXGPT, the cognition performance of vanilla large language models witnesses consistent improvements regardless of the model architecture and size variation. In particular, we boost the open-sourced LLaMA2-70B [53] to achieve comparable performance

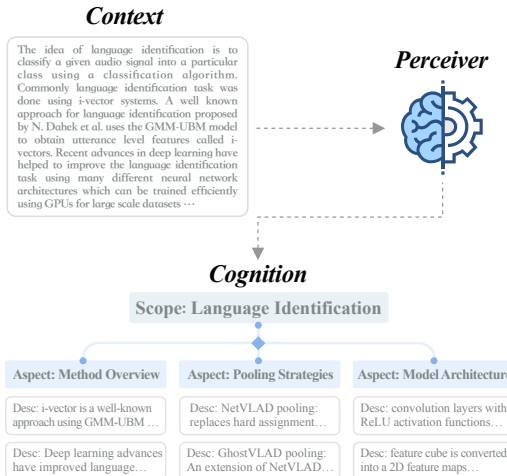

Figure 1: **Structured cognition on sequential contexts**. Humans may easily identify a given passage's topic/scope, break down the text sentences into several aspect points with detailed descriptions, and form a tree-like knowledge structure.

against GPT-3.5-Turbo as a hallucination evaluator, and demonstrate the compatibility with other advanced prompting techniques, such as CoT [59]. We hope this paper can bring new insights to the community on building a more powerful and critical language model with human cognition.

Our contribution can be summarized as follows:

- We propose the concept of structurization, in order to enhance LLM's cognition capability without altering the models themselves.
- We present the feasibility of distilling the structurization ability from giant commercial LLMs into a responsive and affordable StruXGPT-7B model, making our approach practical.
- With structurization, we empirically demonstrate the consistent cognition enhancement for various LLMs across model architecture and size variation on diverse NLP tasks.

## 2   Related Work

**Large language models (LLMs)**. LLMs' *emergent abilities* [58] has recently received extensive attention in the literature [6, 61, 9, 53, 23, 1], which are found closely related to the scaling law [26]. When the scale reaches a certain level, the performance on complex NLP tasks significantly rises due to the superior *in-context learning*, *instruction following*, and *reasoning* abilities [71, 47]. Numerous efforts have been made to boost the model capacity with training and prompting strategies [37, 38, 59, 14], and this paper presents a new perspective, *context structurization*, to encourage LLMs to perceive, recognize, and communicate like humans [55, 71], without altering the model themselves.

**Context augmentation**. Recent studies have proposed several context augmentation methods to enhance LLM's cognition ability [59, 41] on when taking the long-form context (with thousands of tokens) as inputs [3, 36, 30]. Specifically, aspect-based summarization (ABS) and query-based summarization (QBS) [64, 69] are designed to extract important information from lengthy text data as well, but they require pre-defined or user-input aspect/query lists to conduct targeted summarization, and the detailed information will be inevitably lost during the summarization process. In contrast, the

---

[1] https://openai.com/
[2] https://dashscope.aliyun.com/

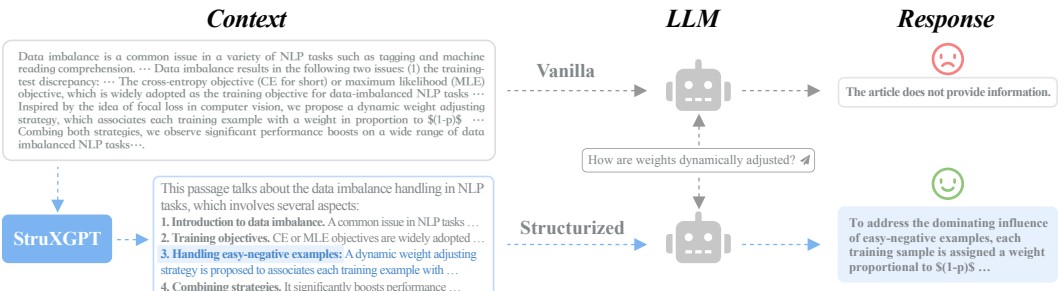

Figure 2: **Framework overview.** When instructed to generate responses based on vanilla long-form and sophisticated contexts, LLMs often lose their focus and give unreliable answers due to their limited cognition capability. In contrast, we *structurize* the vanilla context by using our StruXGPT to identify its main scope and aspect points, facilitating the original LLMs to comprehend the context and generate accurate responses.

paper develops context-wise structurization to highlight the knowledge structure, running a single-turn structurization on the context to enhance LLMs' cognition abilities on a diverse set of NLP tasks.

**Structurization**. In the conventional NLP literature, the term *structured data* usually refers to entity-relation-entity triplets or properties extracted from plain texts, which are utilized to construct knowledge graphs or databases with special data formats or schemas [24, 32]. On the contrary, the *structurization* in this paper does not focus on entity-level information extraction and aggregation. Instead, it suggests reorganizing the input sentences into a three-layer structure based on their inner linguistic relations. Similar to discourse analysis and constituency parsing [10, 27], the main purpose of our structurization is capturing the dependencies and relations of the elements within specific long-form text inputs, so as to enhance LLMs' cognition of the knowledge structure and relations.

**Knowledge Distillation**. In the era of large language model learning, distilling specific knowledge from giant LLMs' outputs has commonly been used to derive a more affordable but still powerful language model [63, 19]. Previous attempts, such as Standford Alpaca [50] and Vicuna [8], show the feasibility of collecting instruction-response pairs from GPT-3.5 or GPT-4 to train a smaller fundamental [46] or domain-specific (*e.g.*, with reasoning [43] or coding [20] skills) language models. Our work also distills the structurization capability from the giant Qwen-Max model to a smaller yet effective StruXGPT-7B model, with extensive evaluations to show the efficiency and efficacy.

## 3 Structurization

The substantial purpose of structurization is to mimic the human cognition process and transform plain, sequential text sentences into a well-organized, hierarchical knowledge structure. Inspired by the linguistic discourse analysis [11, 18], this paper develops a three-layer hierarchy exemplar to present the cognition/knowledge structure, as introduced in Fig. 1:

(1) *Scope* summarizes the topic and boundary of the textual context. It outlines the central issues of knowledge throughout the text and the scope of the discussion that will be covered.

(2) *Aspect* further subdivides the input context into several parts. It presents the aspects or dimensions that must be considered to fully understand the topic and scope.

(3) *Description* is the most specific and detailed layer. It provides in-depth descriptions and analyses to support each aspect of the context scope.

This generic three-layer structure is derived for efficacy and efficiency in dealing with diverse textual inputs. There might be some elaborated structures (such as a knowledge mindmap [60]) to better deconstruct specific text sources, but the difficulty and complexity of defining, extracting, and utilizing those structures to aid in practical problems are dramatically increased. We leave this exploration in our future work.

Next, we will present how to implement effective structurization with minimal cost in Sec. 3.1, and demonstrate the utilization of the structurization results to enhance LLMs' cognition abilities in different downstream tasks in Sec. 3.2.

## 3.1 Efficient Implementation of Structurization

We have explored two approaches to execute the structurization process by leveraging the extraordinary capability of large language models: few-shot prompting on commercial LLMs and direct instructing our developed StruXGPT models.

**Few-shot commercial LLM.** In the initial stage, we use two in-context examples to query the commercial Qwen-Max model to structurize the input corpus, since it shows promising instruction-following and textual analyzing capability with over 200B parameters [3]. Here is a simplified example of prompting structurization in Fig. 3, and the full template is displayed in Appendix E.1.

However, commercial LLMs are usually slow and expensive, and sending user data to LLM APIs may cause privacy and security problems due to information leakage. Thus, we train a smaller 7b-parameter model (*e.g.*, LLaMA2 [53] or Qwen [2]) to inherit the structurization ability from giant commercial LLMs, which can be deployed locally for efficiency and privacy.

**Fine-tuned StruXGPT.** Our tailored model is named by StruXGPT, where *Stru* is the abbreviation of structurization, and *X* implies we do not specify the model architecture. We carefully curate 22,547 raw data pieces from Wikipedia[4] and CAMEL-AI dataset [31] to ensure diversity, and collect the structurized results from Qwen-Max to train our StruXGPT via supervised fine-tuning (SFT). From the collected samples, 200

```
Given a sequential statement, you are supposed to identify:

## Statement's scope:
```[generally a noun phrase]```

## Statement's main aspects and corresponding descriptions:
```
1. [the first aspect of the statement]
    1.1 [a descriptive sentence]
    1.3 [another descriptive sentence]
2. [the second aspect of the statement]
    2.1 [a descriptive sentence]
3. [another aspect of the statement]
```

### Input:
{input_statement}

### Output:
```

Figure 3: Prompt template for structurization.

are utilized for evaluation (including human verification), and the remaining training samples are adopted to distill the structurization capability from Qwen-Max to our StruXGPT-7B. It is practical since the structurization only relies on fundamental syntactic understanding and processing ability, which has already been learned from the large-scale corpus. We merely teach the 7B-parameter models how to reorganize the input text via SFT, without introducing new memorizing or creative overloads. In addition, the ultimate StruXGPT does not require few-shot examples, and it reduces the input lengths for further efficiency. The training details are described in Appendix A.1.

## 3.2 Effective Utilization of Structurization

The identified knowledge structure (*i.e.*, the scope, aspect, and description hierarchy) from the raw context is initially parsed in JSON format and unsuitable for direct inputs to handle massive lengthy elements. Therefore, we use a unified template to transform the structured data back into natural language sentences as models' inputs to fit their intrinsic processing patterns, as LLMs are pre-trained and aligned with mostly natural language data. Concurrently, to preserve and highlight the knowledge structure, we harness specific linguistic markers to signal hierarchy and relationships among concept elements, such as numbered lists for order, bullet points for categorization, and indentation to depict nesting levels of information. Fig. 4 showcases some typical examples.

The first row of Fig. 4 provides a unified template to transform structurization results into natural languages. The top-level hierarchy *scope* is presented as a standalone sentence, serving as the introduction to the structured context and highlighted with bold markers. Subsequently, the secondary *aspect* are organized with numerical markers and bolded, attaching with its corresponding tertiary *descriptions* through subclauses or separate sentences. This method not only signifies the rank and relation of each piece of information relative to others but also provides clear, navigable paths for the LLMs to follow and process the information efficiently (*e.g.*, when examining the long-form comprehension capability). Moreover, the second row of Fig. 4 introduces another variation for transformation, where each *description* elements are further broken down and enumerated for the delineation of fine-grained details, making it easier for language models to discern and retain specific nuances associated with each *aspect* (*e.g.*, when examining the hallucination detection capability).

---

[3] https://rank.opencompass.org.cn/leaderboard-llm-v2
[4] We use the 20231020-en dump from https://dumps.wikimedia.org/.

**This passage talks about {Scope}:**
1. **{Aspect 1}**: {Desc 1.1}. {Desc 1.2}.
2. **{Aspect 2}**: {Desc 2.1}. {Desc 2.2}.
3. **{Aspect 3}**: …
4. …

**This passage talks about the PrivacyQA dataset and its characteristics:**
**1. Introduction to Privacy Policies**. Privacy policies are legal documents that disclose how companies gather, use, share, and manage user data…
**2. Privacy Policy Misuse and Lack of Awareness**. Lack of awareness …
**3. PrivacyQA Dataset**. Motivated by the need for quick identification …
**4. Data Collection Methodology**. PrivacyQA comprises 35 mobile application privacy policies collected from the Google Play Store…

**{Scope} can be deconstructed as:**
1. **{Aspect 1}**
   - {Desc 1.1}.
   - {Desc 1.2}.
2. **{Aspect 2}**
   - {Desc 2.1}.
3. …

**The life and career of director George Cukor can be deconstructed as**:
**1. Early life and background**
   - Cukor was born on the Lower East Side of Manhattan, the younger child and only son of…
   - His parents selected his middle name in honor of Spanish–American War hero George…
**2. Career milestones**
   - Cukor won the Academy Award for Best Director for "My Fair Lady" (1964), which was …
   - He continued to work into the 1980s….

*Templates*     *Examples*

Figure 4: **Left**: templates to transform structurization results into natural languages, with special linguistic markers to preserve and highlight the extracted knowledge structure. **Right**: transformed context examples with clear information structure for long-form reading comprehension (upper) and hallucination detection (lower) tasks.

After transformation, the linguistic input retains its structured knowledge through systematic cues but is presented in a comprehensible manner for LLMs. This not only facilitates an enhanced understanding and interaction with complex data but also enables the models to leverage their existing natural language capabilities to generate more accurate and contextually relevant responses.

# 4 Experiments

In this section, we conduct extensive experiments on a series of downstream NLP tasks to comprehensively demonstrate the efficacy of structurization. We hope the results can bring new insights to enhancing LLM's cognition via structurization, regardless of model architecture and size variation.

Three representative natural language understanding and processing tasks are investigated, including context-based question-answering Sec. 4.1, exhaustive hallucination evaluation Sec. 4.2, and passage-level dense retrieval Sec. 4.3. When instructing tested LLMs to perform target tasks, we merely structurize the vanilla textual inputs, transform the results back into natural languages, and immediately feed the results to LLMs to make responses. The tested LLMs themselves are not fine-tuned on structurized data corpus. We adopt our StruXGPT-7B model to execute structurization in all experiments in this section. The evaluation and ablation of StruXGPT model are demonstrated in Sec. 4.4, Sec. 4.5, and Sec. 4.6, and more comparison with related augmentation-based methods can be found in Appendix B.3 and Appendix B.5.

## 4.1 Application on Context-based Question-Answering

Question-answering based on a long-form context is an emerging research area within QA, which requires large language models to precisely seek the target information and generate reliable responses to the question [3, 30]. It is an immediate measure of LLM's cognition ability to handle intricate and sophisticated contexts. In this section, we comprehensively evaluate how structurization boosts QA ability on seven datasets from the LongBench benchmark [3] with a variety of LLMs to examine.

**Dataset setup.** LongBench [3] is a multi-task benchmark tailored for long context understanding evaluation, composed of 6 major task categories and 21 different tasks. To focus on the investigation of context structurization, we choose 7 subsets from LongBench across single-document QA, multi-document QA, and synthetic QA tasks in English, and the remaining Chinese subsets or code-orientated tasks are eliminated. Except for the *MultiFieldQA* subset with 150 testing samples, each subset contains 200 pieces of *context-question-answer* triplets to evaluate, resulting in 1,350 samples to test in total. Each subset has a 4K-18K context length on average. If the context length exceeds an LLM's window size, we truncate from the middle of the text and preserve information at the beginning and end, as suggested by LongBench. Detailed dataset description is displayed in Appendix B.2.

Table 1: **Performance on LongBench datasets.** The indicator +*StruXGPT* means the data fed into LLMs is structurized by our StruXGPT-7B, while the evaluated LLMs themselves are unchanged. The results are acquired by LongBench's official protocol. Higher is better.

| Method | SingleDoc QA | | MultiDoc QA | | | Synthetic Tasks | | Average |
|---|---|---|---|---|---|---|---|---|
| | Qasper | MFQA | HpQA | 2Wiki | Musique | PsgCnt | PsgRet | |
| LLaMA2-7B-4k | 19.5 | 34.6 | 30.4 | 27.3 | 10.7 | 2.0 | 9.0 | 19.1 |
| **+StruXGPT (ours)** | **23.1** | **35.9** | **32.7** | **29.9** | **13.4** | **3.0** | **12.0** | **21.4** |
| LLaMA2-13B-4k | 26.9 | 34.5 | 38.9 | 34.4 | 13.9 | 2.0 | 10.0 | 22.9 |
| **+StruXGPT (ours)** | **28.5** | **35.3** | **40.0** | **39.4** | **18.9** | **3.5** | **16.0** | **26.0** |
| Qwen-7B-8k | 19.6 | 34.1 | 20.4 | 12.5 | 7.5 | 2.0 | 15.5 | 15.9 |
| **+StruXGPT (ours)** | **22.3** | **37.0** | **25.4** | **14.7** | **8.2** | **2.5** | **17.5** | **18.2** |
| ChatGLM3-6B-32k | 43.3 | 51.7 | 54.4 | 44.9 | **40.4** | 2.0 | 99.0 | 47.9 |
| **+StruXGPT (ours)** | **44.6** | **52.1** | **57.2** | **47.6** | 40.1 | **4.0** | **99.5** | **49.3** |

**Evaluated models.** Following [3], we evaluate three representative large language models on long context comprehension ability: LLaMA2-7B-4k [53], Qwen-7B-8k [2], and ChatGLM3-6B-32k [67], and extend the larger LLaMA2-13B-4k [53] model. The four LLMs have a relatively similar parameter capacity with different model architectures and window sizes. All the models to examine are pre-trained chat models, and we just employ our StruXGPT to structurize the input contexts to enhance those LLMs' cognition capabilities. To ensure reproducibility and reduce uncertainty, greedy search is adopted during LLM's decoding process when generating responses. The accuracy between models' responses and ground-truth answers is measured by ROUGE-L and F1-score.

**Experimental results.** Tab. 1 suggests structurized contexts bring consistent improvements on almost all 3 tasks and 7 subsets across the model architectures and window sizes. Specifically, structurization leads to relatively greater improvement for the MultiDoc-QA subtask (with a 3% performance gain on average), revealing the potential promotion of LLMs' multi-hop reasoning abilities. Despite the negligible decline on the *Musique* [54] subset (see Appendix B.2 for analysis), the advanced ChatGLM3-6B-32k is also boosted, showing structurization's efficacy for powerful models.

**Comparison with other baselines.** To further evaluate our structurization augmentation, we compare our method against the typical summarization-based methods that also employ LLMs for context augmentation. The results are presented in Appendix B.3, which further demonstrates our superiority in highlighting the knowledge structure without loss of key information.

**Investigating structurization from the attention perspective.** Fig. 5 reveals how structurization can aid LLM's cognition from the attention perspective. In particular, we compare the attention maps for the same tested LLaMA2-7B model with different contexts as input. At the position of the model's first token prediction, we average the attention maps across the 32 attention heads for each layer of LLaMA's last 16 layers [72], and visualize the attention scores in Fig. 5. Specifically, when handling vanilla contexts, LLaMA2-7B loses its focus on the target information of the *experts*. On the contrary, the structurized context clearly presents the content structure of the introduced *PrivacyQA dataset*, and LLaMA2-7B immediately grasps the target aspect and its detailed descriptions of *experts with legal training*. In this way, LLM's cognition capability is successfully enhanced via context structurization.

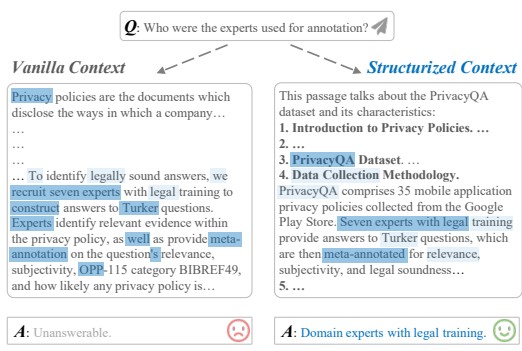

Figure 5: Attention maps on vanilla and structurized contexts for the same LLaMA2-7B. The sample comes from the QAsper subset.

## 4.2 Application on Exhaustive Hallucination Evaluation

Hallucination has raised wide attention in the community [70, 56]. In general, evaluating hallucinations involves verifying atomic claims against supportive materials (*e.g.*, Wikipedia passages [66, 42]), yet even advanced GPT-3.5-Turbo and GPT-4 cannot always accurately make the judge, as LLMs-evaluators often struggle to extract relevant information due to the complexity of passage contexts. We introduce how to improve LLM evaluators' assessing ability by context structurization below.

**Dataset setup.** AttrScore [66] and FactScore [42] datasets are adopted for evaluation. We take the *AttrEval-GenSearch* test set with 245 examples from AttrScore, where each example comprises a statement and its reference passage, and is annotated by *Attributable* (abbreviated as *Attr.*), *Contradictory* (abbreviated as *Contra.*), and *Extrapolatory* (abbreviated as *Extra.*). FactScore collected 4,726 atomic claims/statements of people biographies generated by InstructGPT (abbreviated as InstGPT), 5,426 by ChatGPT, and 5,888 by PerplexityAI (abbreviated as PPLAI). For each input sample, we leverage StruXGPT to structurize the reference to identify its main aspects and detailed descriptions. The numerically ordered structure is preserved, as displayed in Fig. A3, since we explicitly ask the evaluator to check the information along the structure for judgment.

Table 2: **Hallucination Evaluation on AttrScore.**

| Evaluator | *Attr.* | *Contra.* | *Extra.* | **Average** |
|---|---|---|---|---|
| GPT-4 | 87.3 | 45.0 | 89.6 | 74.0 |
| GPT-3.5-Turbo | 61.2 | 20.6 | 53.3 | 45.0 |
| Alpaca-13B | 50.6 | 6.1 | 19.3 | 25.3 |
| Alpaca-7B | 50.7 | 8.6 | 3.6 | 21.0 |
| LLaMA2-7B | 51.5 | 9.1 | 20.1 | 26.9 |
| **+StruXGPT (ours)** | **54.5** | **15.0** | **30.4** | **33.3** |
| LLaMA2-70B | 70.9 | 31.1 | 74.1 | 58.7 |
| **+StruXGPT (ours)** | **75.4** | **35.6** | **78.1** | **63.0** |
| GPT-3.5-1106 | 72.0 | 30.4 | 71.7 | 58.0 |
| **+StruXGPT (ours)** | **77.1** | **31.8** | **77.4** | **62.1** |
| GPT-3.5-1106 + CoT | 76.4 | 35.3 | 74.4 | 62.0 |
| **+StruXGPT (ours)** | **78.9** | **42.9** | **74.5** | **65.4** |

**Evaluated models.** We mainly investigate the open-sourced LLaMA2-7B and LLaMA2-70B models as the LLM-evaluator, and also explore the integration with the close-source GPT-3.5-Turbo-1106 [6] via API access. The main results are presented in Tab. 2. We also report the results with Qwen models on AttrScore and FactScore and the incorporation to more powerful GPT-4 in Appendix B.4.

**Experimental results.** According to Tab. 2, our structurization brings significant enhancements to both LLaMA2-7B and 70B models (for 6.4% and 4.3% on average, respectively). And the powerful GPT-3.5 model also gains 4.1% (from 58.0% to 62.1%). Furthermore, we incorporate the Chain-of-Thought (CoT) technique into the prompt template to clarify the evaluation steps (such as *Carefully read the claim and double-check the answer.*) (denoted as "GPT-3.5-1106 + CoT"). After that, the GPT-3.5 model immediately obtains an improvement of 4.0% (from 58.0% to 62.0%). Consequently, on top of the advanced CoT prompt, our method further enhances the model to a higher accuracy of 65.4% on average, demonstrating our method's compatibility with advanced prompting techniques.

## 4.3 Application on Passage-level Dense Retrieval

Retrieval-augmented generation (RAG) has been empirically validated to significantly bolster LLM's domain knowledge [29, 62], where precise document retrieval plays a vital role. We now investigate how structurization can facilitate dense retrieval for BERT-like masking language models.

**Dataset setup.** BEIR dataset [52] is a popular benchmark for evaluating dense retrievers' zero-shot effectiveness [39, 33], where retrievers are trained on MS MARCO's passage-retrieval training split [44] while directly tested on the BEIR benchmark without finetuning. We focus on our evaluation of the 5 subsets from BEIR *i.e.*, NFCorpus, FiQA, ArguAna, SciDocs, and SciFact.

**Evaluated models.** BERT [13], SimLM [57], and coCondenser [16] are chosen for evaluation, since they achieve state-of-the-art performance on MS MARCO's development split. To convert the structurized passages from our StruXGPT into natural languages, we eliminate the numerical indicators (such as "1.", "1.1", *etc.*) and attach the description statements in the third layer with their aspects. Fig. A3 presents an example. We only structurize passages to enhance retrievers' cognition, and the queries remain unchanged. Following the literature, nDCG@10 results are reported in Tab. 3.

Table 3: **Performance on BEIR subsets.** Retrievers are trained with MS MARCO corpus and directly evaluated on BEIR without fine-tuning.

| Retriever | NFCorpus | FiQA | ArguAna | SciDocs | SciFact | **Average** |
|---|---|---|---|---|---|---|
| BERT | 24.4 | 23.7 | 36.2 | 11.4 | **50.8** | 29.3 |
| **+StruXGPT (ours)** | **24.4** | **24.9** | **40.0** | **11.4** | 50.7 | **30.3** |
| SimLM | 22.2 | 17.3 | 34.2 | 11.7 | 48.2 | 26.7 |
| **+StruXGPT (ours)** | **22.9** | **19.8** | **34.6** | **11.7** | **52.7** | **28.3** |
| coCondenser | 28.2 | 22.8 | 40.5 | 12.8 | 55.6 | 32.0 |
| **+StruXGPT (ours)** | **28.8** | **23.5** | **43.4** | **13.1** | **56.8** | **33.1** |

**Experimental results.** Structurization boosts all three retrievers on most subsets, yielding a maximum performance improvement of 4.5% on SciFact for SimLM. The results suggest that structurization not only augments decoder-only generative LLMs with explosive parameters (at least 7B), but also benefits encoder-decoder masked language models with constrained parameters (around 110M). It implies that the patterning of linguistic and semantic structurization may be a fundamental mechanism for enhancing language models, transcending distinctions in their architectural design and scale.

## 4.4 Evaluation of the Structurization Approach Itself

In this section, we assess various structurization methods through exhaustive experiments on five approaches, including prompting Qwen-max with few-shot exemplars (serves as our teacher model), few-shot Qwen-7B and LLaMA2-7B pre-trained chat models, and our fine-tuned Qwen-7B and LLaMA2-7B (student) models. As introduced in Sec. 3.1, we use 200 validation cases and have the five models generate 1,000 structurized outputs for analysis.

A good structurization should effectively deconstruct the vanilla input text to clearly identify its knowledge structure, so as to facilitate LLM's cognition. The resulting content should be faithful to the original texts, neither dismiss the factual information nor fabricate statements or opinions that do not exist. To this end, we revise four evaluation metrics to investigate the efficacy of different structurization approaches, and the results are displayed in Tab. 4.

Table 4: **Comprehensive comparison on structurization approaches.**

| Approach | Model | LexicalEval | | HumanEval | | | AppEval | SemEval |
|---|---|---|---|---|---|---|---|---|
| | | recall? | precision? | completeness↑ | factuality↑ | anti-hallu↑ | Δ↑ | bertscore↑ |
| Few-shot | Qwen-max | 0.63 | 0.68 | **4.58** | **4.49** | **4.57** | +3.3 | **0.31** |
| | Qwen-7B | 0.56 | 0.67 | 3.77 | 3.67 | 3.96 | -0.1 | 0.22 |
| | LLaMA2-7B | 0.61 | 0.72 | 4.09 | 3.98 | 4.12 | +0.3 | 0.24 |
| Fine-tuned | StruXGPT-7B-Q | 0.63 | 0.67 | **4.41** | 4.36 | **4.48** | +3.6 | **0.31** |
| | StruXGPT-7B-L | 0.61 | 0.66 | 4.37 | **4.38** | 4.36 | +2.8 | 0.30 |

**Lexical evaluation (LexicalEval).** We first leverage the widely-used ROUGE-L [34, 35, 25] to assess recall and precision between structured content and original text. However, lexical metrics from the methods, ranging from 0.6 to 0.7, inadequately reflect structurization quality where LLMs will paraphrase the words but lexical scores miss the semantic consistency. For instance, a statement pair "They adopt ROUGE" and "ROUGE is adopted" only receives a 0.33 f1-score for ROUGE-L. Therefore, a crucial human evaluation is developed to obtain a trustworthy conclusion.

**Human evaluation (HumanEval).** We recruited 17 well-trained natural language annotators from the PAI-iTAG platform[5] to evaluate the structurization quality on a 0-5 scale across three dimensions: completeness (ensuring no loss of information from the original text), factuality (accurate three-layer deconstruction), and anti-hallucination (avoiding fabricated content). As annotating structurization

---

[5]An open platform at `https://www.aliyun.com/product/bigdata/learn/itag`

only involves linguistic and syntactic level judgments, annotators do not need professional expertise to check the information of a given text itself. The detailed evaluation criteria are displayed in Appendix A.3, and we report the labeling results in Tab. 4.

The commercial Qwen-max shows a promising instruction-following and in-context learning ability, generally scoring 4.5 at the three dimensions. However, the pre-trained 7B models from Qwen and LLaMA2 immediately decline the scores to below 4.0, as they struggle to understand the instruction to build the three-layer structure and tend to hallucinate responses due to the limited model capacity. More structurization examples can be found in Appendix E.2. Notably, the fine-tuned StruXGPT-7B-Q(wen) and StruXGPT-7B-L(LaMA2) both obtain a 4.35 - 4.45 score on average. They inherit 97% of structurization capability from the Qwen-max teacher model, evidencing the effectiveness of training a specialized 7B-parameter model for structurization to aid in efficiency and privacy.

**Evaluation with downstream application (AppEval).** Since our main motivation is to utilize structurization to enhance LLM's cognition, a further evaluation is conducted to investigate those different structurization methods. Specifically, we compare how much improvement ($\Delta$) those methods can bring to downstream natural language processing applications. On the Qasper subset [12] from LongBench [3], we instruct an independent LLaMA2-7B chat model for reading comprehension with long-context structurized by different approaches. LLaMA2-7B receives a 19.6 F1-score for QA accuracy when taking the vanilla context as input, which serves as the baseline performance.

The evaluation results are displayed in Tab. 4, which are consistent with the human evaluation presented above. In particular, the few-shot Qwen-max achieves an over 3% improvement in answer quality, while the pre-trained 7B-parameter chat models fail to generate validated structurizations. Meanwhile, our fine-tuned StruXGPT-7B-LLaMA and StruXGPT-7B-Qwen models bring comparable enhancements against the few-shot Qwen-max, emphasizing the efficacy of distilling the structurization ability from giant teacher models to a more responsive and affordable student model.

**Evaluation with semantic embedding (SemEval).** At last, we explore the structurization quality evaluation in the semantic embedding perspective, as a supplementary to lexical evaluation. Following Zhang et al. [68], we calculate the semantic similarity between original and structurized contents with the embedding similarities, and the results in Tab. 4 show consistent measure against HumanEval and AppEval with a much lower cost. Hence, BERTScore [68] can be further leveraged as an effective and efficient quality-assessment tool for training-data filtering and structurization quality evaluation to derive a better StruXGPT model. Sec. 4.6 presents some preliminary investigations.

Through the comprehensive evaluation of three protocols, we demonstrate the feasibility and efficacy of training a specialized StruXGPT. It is more resource-friendly to deploy for efficiency and privacy, meanwhile inheriting 97% of the ability from giant teacher models to perform structurization.

## 4.5 Ablation Studies on StruXGPT's Establishment

This section studies two major factors of training a StruXGPT model: data quality and model capacity.

**Using two few-shot examples is sufficient to collect high-quality training data.** In this work, we choose 2 in-context examples to prompt commercial LLMs (as a teacher) to generate data pairs of raw/structurized texts to train our StruXGPT-7B model (as a student). We think it is enough for teacher models to understand the structurization process and generate valid training samples, as the 2 examples respectively describe the 2 most common types of real-world text (*i.e.*, with/without existing indicators like "1.", "2.", etc), which is displayed in Appendix E.2.

To further verify it, we investigate the number of in-context examples with two evaluation protocols (as in Tab. 4): AppEval (an improvement on Qasper subset with context structurization) and BERTScore (semantic similarity with raw and structurized texts in the validation set). We also report the error rate when parsing structurization results from the teacher model's outputs (denoted as "FormatError").

According to Tab. 5, 1-shot is apparently insufficient to illustrate structurization, while 2- and 3-shot achieve comparable structurization quality under AppEval and BERTScore. Notably, 3-shot receives a 2% lower FormatError than 2-shot, in trade for the increased inference cost (because of increased few-shot samples). We argue that the 2% gap (around 400 samples) does not make a difference for the final StruXGPT training, which can be verified in Appendix A.2. Therefore, we recommend users to apply 3- or even more shots when prompting teacher LLMs if available, otherwise 2-shot is also a good choice to balance the inference cost and structurization quality.

| Table 5: **Number of few-shot examples.** | | | |
|---|---|---|---|
| nShot | AppEval | BERTScore | FormatError |
| 1-shot | +1.8 | 0.282 | 25.4% |
| 2-shot | +3.2 | **0.308** | 7.4% |
| 3-shot | **+3.3** | 0.302 | **5.5%** |

| Table 6: **Parameter capacity of StruXGPT.** | | | |
|---|---|---|---|
| StruXGPT | AppEval | BERTScore | FormatError |
| Qwen-1.8B | +2.7 | 0.299 | 5.0% |
| Qwen-7B | +3.6 | 0.313 | **0.0%** |
| Qwen-14B | **+3.8** | **0.323** | **0.0%** |

**StruXGPT-7B balances parameter capacity and structurization quality.** As Qwen [2] provides a series of models varying sizes, in Tab. 6, we implemented StruXGPT on Qwen-1.8B/7B/14B respectively to investigate the relationship between model capacity and structurization quality.

Compared with the 7B model capacity, the smaller 1.8B model, despite its positive enhancement on downstream applications, shows slight inferiority in both AppEval (+2.7 *v.s.* +3.6) and BERTScore (0.299 *v.s.* 0.313), and presents 5% error rate when parsing structurization results. On the other hand, the 14B model brings further improvement to BERTScore, (the structurization content is relatively more faithful to original texts), but the boost on AppEval is insignificant. Hence, the 7B model is a good trade-off between model capacity (training/inference efficiency) and structurization quality.

### 4.6 Utilization of Structurization Quality Assessment

As the structurization quality simultaneously influences StruXGPT's training performance and application improvements, this section investigates the quality assessment tool BERTScore (as discussed in Sec. 4.2) on StruXGPT's training and inference stages, respectively.

On one hand, as the training data quality determines StruXGPT's upper bound, we statistic the BERTScore of the 22K training entries, and around 94.45% raw/structurized text pairs present positive scores (normalized by the baseline score of 0.83, and a positive BERTScore presents a benign similarity), demonstrating the high quality of our training data. Consequently, we eliminated around 5% of data with negative scores and trained another StruXGPT model, and the results in Tab. 7 indicate this part of data does not affect the final performance. According to Appendix A.2, data quantity may play a vital role in further improving our StruXGPT.

| Table 7: **Training data filtering.** | | |
|---|---|---|
| Training Data | AppEval | BERTScore |
| vanilla | **+3.6** | 0.313 |
| filtered | +3.4 | **0.316** |

| Table 8: **Inference results filtering.** | | |
|---|---|---|
| Context | Declined Ratio | Overall Enhance |
| vanilla | 3.5% | +3.6 |
| filtered | **3.0%** | **+3.7** |

On the other hand, since poor structurization results can lead to suboptimal application performance, we statistic the enhancement variance in the Qasper subset from LongBench [3] in Tab. 8 for structurized context inputs. Our method merely causes degradation on 3.5% of samples (with relatively lower structurization quality), but ultimately receives a +3.6 improvement overall test samples. Furthermore, we filter out the structurization results with a low BERTScore (*e.g.*, <0.05) and take back the original context as input. In this way, the degradation can be alleviated (from 3.5% to 3.0%), further improving the final enhancement to +3.7. The lower bound of our StruXGPT can thus be ensured.

## 5 Conclusion and Discussion

This paper presents a novel concept of context *structurization* to enhance LLM's cognition capability. Extensive evaluations of various representative NLP tasks reveal the consistent enhancement across language model's architectural designs and capacity scales. We demonstrate the feasibility of distilling the structurization ability from giant commercial LLMs into a responsive and private StruXGPT-7B model, addressing the practicality problem. The limitation and future work are discussed in Appendix C. We hope this paper can bring new insights into how to build a more powerful and reliable language model with human cognition and intelligence.

## Acknowledgments and Disclosure of Funding

This work was supported in part by the Fundamental Research Funds for the Central Universities, in part by Alibaba Cloud through the Research Intern Program, and in part by Zhejiang Provincial Natural Science Foundation of China under Grant No. LDT23F01013F01.

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

# A  Implementation Details of Structurization

## A.1  Implementation Details

The data to train our StruXGPT is carefully curated from two main aspects. First, to ensure diversity, we randomly sample 15,000 passages from Wikipedia across various domains, whose token length ranges from 1K to 3K. Then, we also supply round 15,000 statement passages from the domain-specific model- and human-generated content, including the filtered CAMEL-AI [31], FiQA [40], and MedQuad [4] datasets. With Qwen-max API, we collect the structurized results and eliminate the failed cases (usually due to internet inaccessibility and unexpected output format), resulting in 22,547 pieces of training data in total to derive our StruXGPT, and 200 test samples for evaluation.

StruXGPT is built upon LLaMA2- or Qwen-7B-chat models, which had been aligned to humans with promising instruction-following capabilities with their prior knowledge. The 22,547 pieces of input-output pairs are utilized to distill the structurization ability to StruXGPT via supervised fine-tuning (SFT). Specifically, StruXGPT is trained with a constant learning rate of $5 \times 10^{-6}$ for LLaMA and $1 \times 10^{-5}$ for Qwen for 1 epoch. The batch size is 128, and other hyper-parameters follow the default settings from Touvron et al. [53] and Bai et al. [2]. The training is resource-friendly, which can be done on 8 NVIDIA V100 (16G) GPUs for 3.5 hours.

For all the inference experiments, we leverage 1-2 NVIDIA A100-80G GPUs for model deployment.

## A.2  Additional Ablation Studies on Training Data

To further investigate the training sample size and its impact, we have conducted additional ablation studies on StruXGPT's training process, and report the question-answering capability enhancement for the same LLaMA2-7B-Chat model on LongBench's Qasper subset, as introduced in Sec. 4.4.

Our results in Tab. A1 reveal an interesting pattern: as the training sample increases, StruXGPT first causes a decline in the LLaMA2-7B model's QA performance, and then brings stable enhancement with larger training data (over 10K samples). We attribute this to the fact that we train StruXGPT for only one epoch in each experiment to prevent overfitting. Consequently, models trained with too few samples often exhibit subpar structurization quality, commonly resulting in information loss or hallucination, which adversely affects downstream performance.

Table A1: Ablation on training samples.

| Training Samples | Enhancement |
|:---:|:---:|
| 2K | -0.6 |
| 5K | -1.3 |
| 10K | +2.6 |
| 15K | +3.3 |
| **22K** | **+3.6** |

Specifically, for the model trained with 2K samples, the generated outputs frequently contain incorrect formatting and struggle to parse the three-tier structure effectively. In these cases, we resort to using the original text as a fallback input for downstream LLMs, which accounts for the somewhat lower performance drop (-0.6%) compared to the model trained on 5K samples (-1.3%).

## A.3  Human Evaluation

**Criteria.** A good structurization should effectively deconstruct the vanilla input text to clearly identify its knowledge structure, and the structurized results should be faithful to the original texts, neither dismissing the factual information nor fabricating statements or opinions that do not exist. Therefore, we devise three dimensions to evaluate the structurization quality, and each of the dimensions can be scored from 0 to 5 (higher is better). The detailed criteria are displayed in Fig. A1, and the annotating screenshot is provided in Fig. A2.

**Instruction.** The full annotation instruction is displayed below:

> Assess the quality of structured outputs produced by different sources, and evaluate whether the structured output is sufficiently "good" by using a score ranging from 0 to 5. To clarify what constitutes a "good" structured result, please consider and score the following aspects, with each aspect also scoring from 0 to 5.
> Scoring Criteria: Please see Fig. A1.

**Payment.** The payment for annotators is 2,000¥ in total, which is higher than the minimum wage in our country.

| Score | Completeness | Factuality | Anti-Hallucination |
|---|---|---|---|
| 5 | The core content information is largely retained with the omission of 2-3 descriptive clauses or adverbials permitted. | The structure is clear and standard with appropriate granularity as expected. | The content is consistent to the original text, with no fabrications/hallucinations. |
| 4 | There is an omission of 20% of sentences or the absence of introductory and concluding overview/summary paragraphs. | The hierarchical structure is retained but with minor formatting issues, such as extraneous spaces or line breaks; granularity is generally suitable, with occasional instances of a subsection containing only one description or exceeding seven descriptions. | The overall content is faithful, but it includes a few expanded sentences not present in the original text. |
| 3 | There exist considerable omissions of sentences or paragraphs are present, or each paragraph lacks 40% of its sentences. | Poor structure: lack of aspect/description numbering, excessively long aspect name, or overly short description sentences (lists of words); Inappropriate granularity: all sentences attached to one aspect, or dividing each sentence into a separate aspect. | Considerable content is fabricated, such as offering additional large sections of ``suggestion'' text. |
| 2 | Extensive information loss is evident, with more than half of the original sentences removed. | The basic structure is preserved, but there is a frequent lack of numbering and non-compliance with hierarchical organization. | Severe repetition or meaningless phrases are present, significantly reducing the quality of the result. |
| 1 | The majority of information is omitted, with only a few key words or sentences preserved. | Chaotic structure: content is organized without any adherence to hierarchical structure. | Essentially nonsensical, with only a few words or phrases retained from the original text. |
| 0 | If the content is empty or completely hallucination, give it a score of 0. | | |

Figure A1: Detailed descriptions for human evaluation criteria on structurization quality.

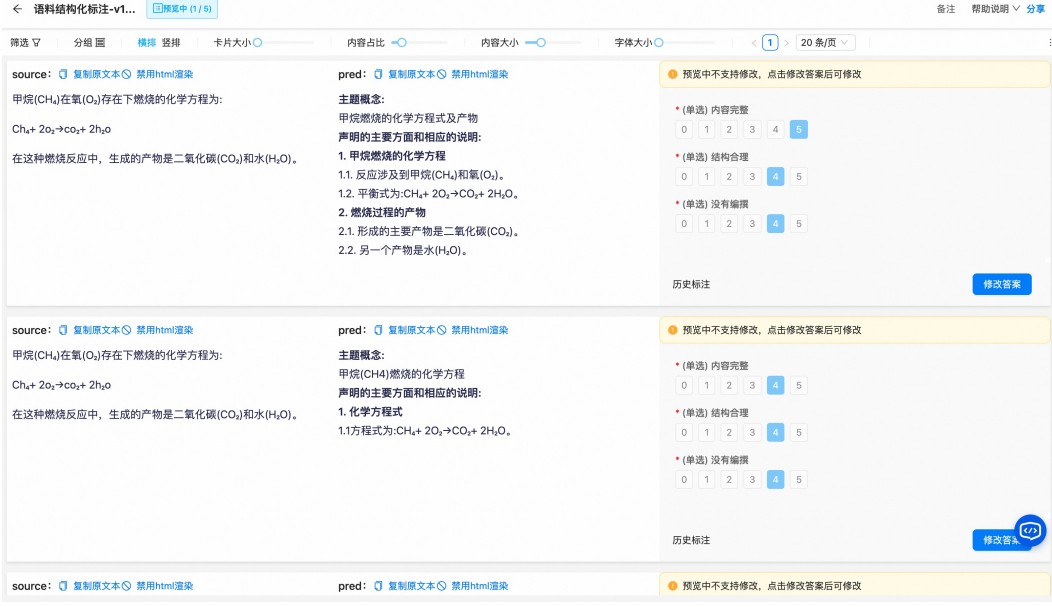

Figure A2: Screenshot for human evaluation.

# B Details for Structurization Applications

## B.1 Structurization Examples

Fig. A3 provides several examples of how to structurize the long-form context for downstream applications, including reading comprehension in Sec. 4.1, hallucination detection in Sec. 4.2, and passage retrieval in Sec. 4.3.

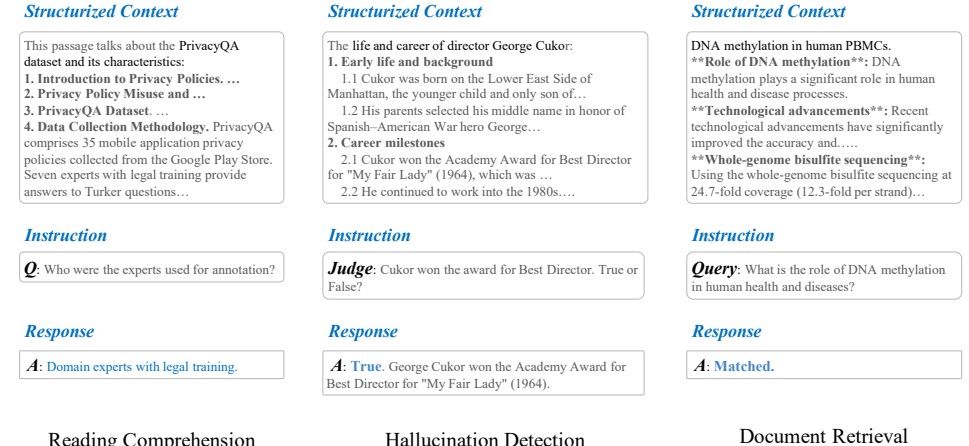

Figure A3: Examples to leverage the structurized results for downstream aookucatuibs

## B.2 Details for Context-based Question-Answering on LongBench

**Detailed dataset description.** To focus on the investigation of context structurization, we choose 7 subsets from LongBench across single-document QA, multi-document QA, and synthetic QA tasks in English, and the remaining Chinese subsets or code-orientated tasks are eliminated:

- **Single-Doc QA.** For single-document QA, we take two subsets from LongBench: (1) *Qasper* [12], featured by question-answering over NLP technical papers and annotated by NLP practitioners; (2) *MultiFieldQA*, manually curated from multiple data sources and annotated by Ph.D. students. *MultiFieldQA* contains 150 Context-Question-Answer triplets to test, and *Qasper* and other adopted subsets include 200 pieces of test samples respectively.

- **Multi-Doc QA.** Multi-document QA requires LLMs to extract and combine information from multiple documents to derive the answer, which is generally more challenging than single-doc QA. We take three multi-hop QA datasets: (1) *HotpotQA* [65], containing 2-hop questions written by native speakers given two related paragraphs; (2) *2WikiMultihopQA* [22], involving up to 5-hop questions synthesized through manually designed templates on Wikipedia passages; and (3) *MuSiQue* [54], carefully composed with up to 4-hop reasoning on an increased number of supporting and distracting context evidence.

- **Synthetic QA.** We employ two extra synthetic tasks to test LLM's long-context handling ability on specific scenarios and patterns: (1) *PassageCount*, requiring models to count the number of unique passages from a shuffled passages pool; and (2) *PassageRetrieval*, randomly choosing one of 30 passages to obtain its summarization with GPT-3.5-Turbo, and asking tested LLMs to determine the original passage with which the summarization is crafted.

**Performance analysis on MuSiQue.** On the LongBench dataset [3], ChatGLM-3-32K [67] receives a slight decline (around 0.1%) on the MuSiQue subset [54] when taking structurized contexts as input. The main reason is that MuSiQue is too complicated. It requires up to 4-hop reasoning on manually paraphrased questions on a large amount of supporting and distracting context evidence. For ChatGLM-3-32K that has already achieved a considerable cognition ability, context structurization in the inference stage is temporally unable to bring significant advances in such a complex scenario. Next, we will delve into developing methodologies for training LLMs to capture the intrinsic structure of context, unlocking structurization's potential in enhancing LLMs on more complicated NLP tasks.

## B.3  Comparison with Aspect- and Query-Based Summarization

Aspect-Based Summarization (ABS) and Query-Based Summarization (QBS) are classical context augmentation methods in the NLP literature, which aims to gradually extract important information from lengthy text data based on pre-defined aspect list or user-input queries. The summary that contains key information will be integrated into context inputs to augment tested LLMs to generate reliable responses, rather than lost in irrelevant sentences in the original lengthy inputs. We take ABS and QBS as the comparative baseline to further evaluate the efficacy of our proposed structurization augmentation strategy.

We investigate ABS and QBS on five document-QA subsets in LongBench [3]. Specifically, as the pre-defined aspect list is unavailable for each passage in LongBench, we turn the traditional ABS to input-agnostic paragraph-based summarization to lengthy paragraphs, where the prompt template is: *Summarize the following text with no more than three sentences. Passage: {text}; Summary: .* For QBS, the prompt template focuses on specific queries: *Summarize the following text to answer the query with no more than three sentences. Query: {query}; Passage: {text}; Summary:.* The *{text}* and *{query}* are placeholders, and the output summaries are respectively concatenated to form the augmented passages as inputs. We report the performance variance of the same test model (*i.e.*, LLaMA2-7B-Chat [53]) when taking vanilla, summarized, and our structurized passages as inputs to answer the same given questions, as displayed below (due to the space limitation, we merely report the averaged enhancement on the five subsets).

Table A2: **Comparison with ABS and QBS on subsets of LongBench.**

| DataAug | Qasper | MultifieldQA | HotpotQA | 2WikiMQA | Misque | **Average** |
|---|---|---|---|---|---|---|
| - | 19.5 | 34.6 | 30.4 | 27.3 | 10.7 | 24.5 |
| ABS | 15.6 | 24.9 | 29.1 | 27.7 | 8.5 | 21.2 |
| QBS | 21.4 | 30.1 | 31.2 | 27.5 | 13.1 | 24.7 |
| Ours | **23.1** | **35.9** | **32.7** | **29.9** | **13.4** | **27.0** |

Based on Tab. A2, we observed that ABS, without guidance from pre-defined aspects lists or user-input queries, failed to preserve critical information, and led to performance declines in four out of the five subsets. Conversely, QBS, with user query guidance, achieved improvements in the Qasper, HotpotQA, and Musique subsets. However, substantial information loss during the summarization process resulted in decreased performance on the MultiFieldQA subset. In contrast, our structured approach delivered consistent improvements across all subsets, demonstrating its efficacy.

In addition, ABS and QBS are applicable for passage-based question-answering tasks but do not extend well to other tasks such as hallucination assessment, where every piece of information counts and should not be summarized. Our approach is task-agnostic, highlighting the generalizability.

## B.4  Additional Experiments for Hallucination Evaluation

**Qwen models as the LLM evaluators.** In Sec. 4.2 in the manuscript, we mainly study structurization with LLaMA2 and GPT-3.5-Turbo on the AttrScore benchmark. For a thorough comparison, this section provides the investigation of employing another series of models, *i.e.*, Qwen-7B and Qwen-72B [2], on AttrScore and FactScore datasets to further demonstrate our method's efficacy. Results are presented in Tab. A3 and Tab. A4, respectively.

For the 7B- and 72B-parameter Qwen evaluators, structurization can immediately improve judgment accuracy on both AttrScore and FactScore datasets. The smaller Qwen-7B model receives a more significant enhancement, along with a nearly 10% improvement in all metrics (except the *Attr.* criteria in Tab. A3) on the two datasets. It indicates smaller LLMs may be more reliant on structurization to enhance their cognitive and information-seeking capabilities than large-scale LLMs. Simultaneously, the larger Qwen-72B evaluator also witnesses a considerable improvement with a 3%-4% increase. In particular, based on the structurized references, Qwen-72B achieves a competitive evaluation performance against GPT-3.5-Turbo on the FactScore benchmark, and even slightly outperforms GPT-4 on the *Contra.* criteria in AttrScore. Those results substantiate the effectiveness of leveraging structurization to further enhance powerful language models with considerable size capacities.

Table A3: **Results on AttrScore evaluation.**

| Method | *Attr.* | *Contra.* | *Extra.* |
|---|---|---|---|
| GPT-4 | 87.3 | 45.0 | 89.6 |
| GPT-3.5-Turbo | 61.2 | 20.6 | 53.3 |
| Alpaca-13B | 50.6 | 6.1 | 19.3 |
| Alpaca-7B | 50.7 | 8.6 | 3.6 |
| Qwen-7B | 60.9 | 11.6 | 34.2 |
| **+Ours** | **61.9** | **25.9** | **44.4** |
| Qwen-72B | 75.8 | 37.7 | 77.6 |
| **+Ours** | **77.8** | **47.5** | **80.2** |

Table A4: **Results on FactScore evaluation.**

| Method | InstGPT | ChatGPT | PPLAI |
|---|---|---|---|
| GPT-4 | - | - | - |
| GPT-3.5-Turbo | 87.5 | 80.2 | 65.8 |
| LLaMA-65B | 54.6 | 42.1 | 36.1 |
| InstLLaMA-7B | 80.1 | 67.1 | 55.1 |
| Qwen-7B | 68.5 | 50.8 | 40.8 |
| **+Ours** | **79.2** | **64.8** | **46.0** |
| Qwen-72B | 86.9 | 75.2 | 60.5 |
| **+Ours** | **89.0** | **79.4** | **61.8** |

**Incorporation with GPT-3.5/4 models and CoT techniques.** In Sec. 4.2, we the popular chain-of-thought (CoT) technique for prompt augmentation on the advanced GPT-3.5-Turbo model on the AttrScore [66] benchmark, where Tab. A5 is a simply copy of the experimental results. Our method presents comparable performance against CoT, and achieves better enhancement after integrating with CoT, illustrating the compatibility and extensibility to more advanced strategies.

Consequently, Tab. A6 further presents the incorporation of our StruXGPT with the more powerful GPT-4 model and the CoT technique, showing consistent benefits for powerful GPT-3.5 and GPT-4 models, either with or without the CoT strategy. The experiments and discussions further validate the effectiveness of our StruXGPT approach.

Table A5: **Integration with GPT-3.5 and CoT.**

| Method | *Attr.* | *Contra.* | *Extra.* | *Avg.* |
|---|---|---|---|---|
| - | 72.0 | 30.4 | 71.7 | 58.0 |
| +Ours | **77.1** | **31.8** | **77.4** | **62.1** |
| +CoT | 76.4 | 35.3 | 74.4 | 62.0 |
| +CoT+Ours | **78.9** | **42.9** | **74.5** | **65.4** |

Table A6: **Integration with GPT-4 and CoT.**

| Method | *Attr.* | *Contra.* | *Extra.* | *Avg.* |
|---|---|---|---|---|
| - | 86.2 | 43.3 | 88.3 | 72.6 |
| +Ours | **87.6** | **48.3** | **89.7** | **75.2** |
| +CoT | **88.8** | 48.9 | 89.7 | 75.8 |
| +CoT+Ours | 88.5 | **52.8** | **90.3** | **77.2** |

## B.5   Comparison with Other Augmentation Methods

Besides the summarization-based and prompting-based augmentation methods (in Appendix B.5 and Appendix B.4 respectively), here we also compare our structurization approach with AdaptLLM [7], which developed several domain-specific LLMs (in BioMedicine, Finance, and Low) via the proposed reading-comprehension training technique. The experimental results are presented in Tab. A7.

Table A7: **Comparison with AdaptLLM on various domain benchmarks.**

| Domain | Subset | Metric | Baseline | AdaptLLM | **Ours** |
|---|---|---|---|---|---|
| Medicine | PubMedQA | Acc | 59.6 | **63.3** | 63.0 |
| Finance | ConvFinQA | EM | 29.2 | **41.5** | 36.5 |
| Law | SCOTUS | mic-F1/mac-F1 | 28.3/10.8 | 30.0/**17.8** | **30.6**/15.6 |
| General | BoolQ | Acc | 55.7 | 53.9 | **58.2** |

Accordingly, our method can also boost the Llama-7b baseline for 3%-7% *without training*, while AdaptLLM requires costly continual training of the baseline model on each domain corpus. In particular, for the general reading comprehension task, AdaptLLM-Fin does not introduce significant boosts, while AdaptLLM-Bio/Law even cause performance drops, which is mainly because AdaptLLM's domain-adaptation tuning will harm the general capability more or less. In contrast, our method does not alter the baseline model, but only structurizes the input context to enhance LLM's cognition

ability on downstream tasks, showing stable and consistent improvements (*e.g.*, a 2.5% increase on BoolQ). Although our final performance is slightly inferior to the domain-specialized AdaptLLM, our generalizability emphasizes the contribution of our work, as we bring consistent enhancement across downstream domains and cause no degradation on general tasks.

# C  Limitation and Future Work

**Training-time aggregation.** This paper mainly investigates structurization during LLM's inference stage. Despite the universal enhancement across language models and NLP tasks, structurization has not yet fully tapped into its potential. Specifically, developing methodologies for training LLMs to capture the intrinsic structure of context and extrinsic correlations among training corpus presents an unresolved challenge in the field. We will delve into this problem in our future work.

**Inference efficiency.** We acknowledge that the introduced inference cost is a common limitation for the methods adopting LLMs for data augmentation, which depends on computing resources and the optimization techniques employed. Here we conduct extra experiments with the competitive augmentation methods (*i.e.*, ABS and QBS in Appendix B.3) for an in-depth comparison on the Long-Bench [3] dataset for question-answering evaluation. Besides, we also investigate the AttrScore [66] dataset for hallucination evaluation to assess the impact of input length and output length, as well as the model size. The inference time, measured in seconds per sample, is calculated on an NVIDIA A100 GPU with vllm [6] acceleration (except for the LLaMA2-70B model, which demands at least two A100 GPUs for deployment).

Table A8: **Time cost analysis for reading comprehension in LongBench.**

| DataAug | Enhancement | Task-Agnostic | Extra Cost | Total Cost |
|---|---|---|---|---|
| - | +0.0 | ✓ | - | 0.7s |
| ABS | -3.3 | ✓ | 2.7s | 3.4s |
| QBS | +0.2 | × | 2.5s | 3.2s |
| **Ours** | **+2.5** | ✓ | 2.9s | 3.6s |
| † Ours + SoT | *+2.5* | ✓ | *1.2s* | *1.9s* |

LongBench contains input passages with an average length of 14K tokens (computed with the LLaMA2 tokenizer), which burdens the cost for all the data augmentation methods. According to Tab. A8, only the task-specific QBS can bring a positive enhancement of 0.2% on average, which is negligible compared with the 2.5% boost by our task-agnostic structurization approach. However, the LLM-based data augmentation methods generally introduce an extra 2.5-2.9 seconds cost. To alleviate this problem, we have recently noticed a new decoding acceleration technique named Selection-of-Thought (SoT) [45], which is also motivated by the aspect-based thinking and writing process of humans and is naturally compatible with our structurization process. With the 2.39× speed up and negligible quality decrease (denoted as "† Ours + SoT"), the extra cost of our method can be further reduced to an acceptable 1.2 seconds, which we believe makes it more practical.

Table A9: **Time cost analysis for hallucination evaluation in AttrScore.**

| LLM-Evaluator | Enhancement | Total Cost | Extra Cost (%) |
|---|---|---|---|
| LLaMA2-7B | +0.0 | 1.4s | - |
| **+Ours** | +6.4 | 3.5s | 150% |
| LLaMA2-70B | +0.0 | 17.6s | - |
| **+Ours** | +4.3 | 19.7s | 12% |

AttrScore requires a relatively longer output length (with a detailed explanation for hallucination evaluation), and the vanilla time cost increases compared to LongBench. According to Tab. A9, our

---

[6]https://github.com/vllm-project/vllm

structurization earns a 6.4% gain on the average hallucination evaluation accuracy (detailed values are shown in Tab. 2), in trade of the $1.5\times$ increased time cost for the vanilla LLaMA2-7B evaluator. As the base evaluator is scaled up to a larger LLaMA2-70B model, our method merely introduces 12% extra cost (2.1s) and still gains a considerable enhancement of 4.3%. Our method showcases more advantages by incorporating larger base models.

**Performance on General Evaluation Benchmarks** As our method focuses on context structurization to enhance LLM's cognition ability, for the tasks where no context is provided, our method may not bring significant enhancement and even get worse. The commonly used MMLU benchmark [21] is a typical scenario, where LLMs are asked to answer questions without context references but requiring their parametric knowledge (learned during large-scale pre-training), and context structurization does not help. If we insist on structurizing the question alone and feeding it into LLM's inputs, the model may be disturbed by the introduced information from StruXGPT and generate wrong answers. As shown in Tab. A10, our method causes a slight 0.1% decrease (measured by OpenCompass protocol[7]) when taking LLaMA2-7B-Chat [53] as the baseline model.

Table A10: **Evaluation on general benchmarks.**

| Model | MMLU | BBH |
|---|---|---|
| LLaMA2-7B-Chat | **45.93** | 30.47 |
| +Ours | 45.84 | **31.30** |

On the other hand, we have also tested another common benchmark, BBH [49], which is designed to evaluate LLMs' reasoning capability when dealing with several logical sentences/statements. In this case, our method can adapt well to highlight the logical relations and boost LLM's reasoning abilities by 0.8%.

In conclusion, we suggest users apply structurization to long-form or logically complex contexts, while taking the original question as inputs when there is no context provided.

# D   Broader Impacts

We discuss the positive and negative societal impacts as follows:

**Positive Societal Impacts**. Through the innovative approach of context structurization, this work significantly enhances the comprehension abilities of Large Language Models (LLMs), leading to more effective and efficient applications in various sectors such as education, healthcare, and customer service. Avoiding the need for larger models, it not only curtails the environmental impact associated with training sophisticated AI systems but also democratizes access to cutting-edge AI technologies. This fosters a broader base for innovation and empowers smaller entities with the tools to contribute meaningfully to technological advancement. Moreover, our method, rooted in human cognitive processes, promises to enrich human-AI collaboration, paving the way for solving complex societal issues by leveraging AI's improved problem-solving capabilities.

**Negative Societal Impacts**. On the flip side, the advances made through structurization bear potential risks, including the augmentation of disinformation campaigns and the creation of more sophisticated deepfakes, which could undermine trust in digital content. The enhanced capabilities of LLMs might also be co-opted for intrusive surveillance, raising substantial privacy and ethical dilemmas. Furthermore, the inherent biases in training data could be deepened, potentially automating and perpetuating social inequalities. The concentration of advanced AI technologies in the hands of a few could limit competition and place immense power over information dissemination with those entities, while increasing reliance on AI for critical decision-making could inadvertently erode human analytical skills over time. To mitigate these concerns, it is essential to establish rigorous ethical standards and robust oversight mechanisms that govern the deployment of LLMs, ensuring transparency and accountability in their use. Furthermore, active research and the implementation of bias detection, correction frameworks, and the development of media literacy programs are crucial

---

[7]https://github.com/open-compass/opencompass

steps to preemptively address the potential misuse, privacy violations, and societal impact of these advanced AI systems.

# E   Prompt Examples for Structurization

## E.1   Prompt Templates

Fig. A4 displays the few-shot prompt template to query giant commercial LLMs to execute structurization. We first introduce the instruction for structurization (including the task definition and output formats), then provide two representative examples to further illustrate the process. The first example in Fig. A5 tells the model to focus on the existing numerical or enumeration indicators to assist in constructing the aspect-description structure. The second example in Fig. A6 teaches the model to automatically summarize the aspects and attach their corresponding descriptions from the raw text sequences. Finally, in Fig. A4 we ask commercial LLMs to structurize the $input\_statement$ from user input as expected.

To ensure completed and faithful structurization on long-form input statements (usually with thousands of tokens), we provide two detailed exemplars in Fig. A5 and Fig. A6 for commercial LLMs' few-shot learning. However, these two exemplars take a considerable context length, which may even confuse the distilled StruXGPT-7B with a smaller size capacity. We thus remove the examples in StruXGPT-7B's prompt template, as StruXGPT-7B has learned how to perform structurization as desired and the functionality of those examples is negligible. Specifically, the $example\_1$ and $example\_2$ (with their introductions) are removed for both efficiency and efficacy.

## E.2   Structurization Examples

In this section, we present several structurization examples from different models as discussed in Sec. 3 in the manuscript. Given an input statement about *Facebook's stock*, Fig. A7 shows that Qwen-max produces a factual and faithful structurization result with the help of few-shot exemplars. It clearly identifies the main aspects of the original statement, and attaches the corresponding descriptions into each aspect. On the contrary, Fig. A8 indicates the smaller pre-trained 7B-size Qwen and LLaMA2 models failed to follow the instructions to produce qualified structurizations. There exists a fabricated "Note" content in Qwen-7B's response. Meanwhile, LLaMA2-7B is even unable to follow the desired format, and information like "Facebook doesn't sell anything tangible" is missing. Fig. A9 demonstrate the effectiveness of our fine-tuned 7B-parameter StruXGPT model, which is able to structurize the input statement as expected, with comparable performance to the giant Qwen-max model with over 200B-size.

```
You are a helpful NLP assistant.

Help me rephrase a given statement to identify:

(1) a summary of the statement's scope, which should generally be a noun phrase with a few words.

(2) a list of main aspects on which the statement is discussing. Each main aspect should be a noun or noun phrase.
The aspect list should be precise and concise to conclude the statement, and the number of aspects should be limited.

(3) an enumeration of descriptive sentences regarding each aspect above, which display the details of those aspects.
Each description sentence must be completed and faithful to the original statement. You should NOT remove any
descriptive segment in this layer.

Given an original statement, the rephrased structure should strictly follow this format:

## Statement's scope:
```[generally a noun phrase]```

## Statement's main aspects and corresponding descriptions:
```
1. [the first aspect of the statement]
    1.1 [a descriptive sentence corresponding to this aspect]
    1.2 [a descriptive sentence corresponding to this aspect]
    1.3 [another descriptive sentence, if necessary]
2. [the second aspect of the statement]
    2.1 [a descriptive sentence corresponding to this aspect]
    2.2 [another descriptive sentence, if necessary]
3. [another aspect of the statement, if necessary]
```

Here is an example to illustrate how to rephrase an input statement as the desired structure. Pay attention to
numerical or enumeration indicators, like '1. ', '2. ', '(1) ', '(2) ', '- ', etc.

{example_1}

Here is another example to illustrate how to rephrase an input statement as the desired structure. When there are no
explicit words indicating the statement's scope and main aspects, please use a few words to precisely summarize the
scope as well as the main aspects. Then you may carefully attach the descriptive sentences to each main aspect.

{example_2}

Now summarize the "scope" of the following statement with a few words, and then rephrase the input statement to its
"main aspects" and "corresponding descriptions" in the numerically ordered format strictly.
Note that the aspect list should be precise and concise to conclude the statement, and the number of aspects should
be limited.
Each description sentence must be completed and faithful to the original statement, and you should NOT remove any
descriptive segment in this layer.

# Input:
{input_statement}

# Output:
```

Figure A4: **Prompt template for few-shot structurization.**

```
# Input:
```
Comprehensive prevention measures for malignant diseases in the rice seedling stage are as follows:
1. Choose disease-free seeds. Do not leave seeds in diseased fields and nearby rice fields. Choose healthy rice and
eliminate diseased, dead, and injured rice.
2. Seed disinfection. Before sowing, soak the seeds with 25% 100g (Xibok) EC 3000 times liquid for 1 to 2 days, or
take 20 grams of 17% Dexinqing wettable powder for every 6 kilograms of rice seeds. Soak the seeds in 8 kg of water
for 60 hours.
3. Deal with diseased rice straw. Do not cover germinated or dry seedlings with diseased straw.
```

# Output:

## Statement's scope:
```The comprehensive prevention measures for malignant diseases in rice seedling```

## Statement's main aspects and corresponding descriptions:
```
1. Choose disease-free seeds
    1.1 Do not leave seeds in diseased fields and nearby rice fields.
    1.2 Choose healthy rice and eliminate diseased, dead, and injured rice.
2. Seed disinfection
    2.1 Before sowing, soak the seeds with 25% 100g (Xibok) EC 3000 times liquid for 1 to 2 days.
    2.2 An alternative is taking 20 grams of 17% Dexinqing wettable powder for every 6 kilograms of rice seeds.
    2.3 Soak the seeds in 8 kg of water for 60 hours.
3. Deal with diseased rice straw
    3.1 Do not cover germinated or dry seedlings with diseased straw.
```
```

Figure A5: **The first example for few-shot structurization.**

```
# Input:
```

The water absorption curve of rice after soaking is an unimodal curve. The inflection point of the curve is the peak
period of water absorption. The relationship between rice water absorption and time is non-linear and can be
expressed by the following formula: a*t+b=c, where a, b, and c are constants. Under different humidity conditions,
the change in the water absorption rate of rice with time is basically similar, that is, the water absorption rate is
between 0 and the point d inflection, and the rate of change accelerates over time. After the inflection point, the
increase in water absorption gradually stabilizes. The changing rules of water absorption and moisture content of
rice are similar, but under different humidity conditions, the relationship between water absorption of rice and
moisture content is different. When the moisture content is low, the water absorption of rice increases as the
moisture content increases. When the moisture content is high, the increase in water absorption of rice gradually
stabilizes. There are three obvious steps for rice seeds to absorb water: First, at the beginning of water absorption,
the water content of the seeds gradually increases, and the water absorption rate slowly increases. Second, during
the peak water absorption period, the water absorption rate increases rapidly. Third, in the later stage of water
absorption, the water content of seeds slowly increases, and the water absorption rate increases. The water
absorption rate of rice seeds is closely related to temperature. In general, water absorption increases as
temperature increases. The relationship between water absorption and temperature can be expressed as: unsaturated
water absorption (%) = 14.289T-10.719 (where T is temperature)
```

```
# Output:
```

```
## Statement's scope:
```characteristics of soaking rice seeds to absorb sufficient water```
```

```
## Statement's main aspects and corresponding descriptions:
```
1. The water absorption curve
    1.1 The water absorption curve of rice after soaking is an unimodal curve.
    1.2 The inflection point is the peak period of water absorption.
    1.3 The relationship between rice water absorption and time is non-linear, which can be expressed by a*t+b=c,
where a, b, and c are constants.
2. Rice water absorption as time changes
    2.1 Under different humidity conditions, the change in the water absorption rate of rice with time is basically
similar.
    2.2 The water absorption rate is between 0 and the point d inflection.
    3.3 The rate of change accelerates over time
    2.3 After the inflection point, the increase in water absorption gradually stabilizes.
3. Relationship between rice water absorption and moisture content
    3.1 The changing rules of water absorption and moisture content of rice are similar
    3.2 Under different humidity conditions, the relationship between water absorption of rice and moisture content
is different.
    3.3 When the moisture content is low, the water absorption of rice increases as the moisture content increases.
    3.4 When the moisture content is high, the increase in water absorption of rice gradually stabilizes.
4. Three stages for rice seeds to absorb water
    4.1 At the beginning of water absorption, the water content of the seeds gradually increases, and the water
absorption rate slowly increases.
    4.2 During the peak water absorption period, the water absorption rate increases rapidly.
    4.3 In the later stage of water absorption, the water content of seeds slowly increases, and the water absorption
rate increases.
5. Relationship between water absorption rate and temperature
    5.1 The water absorption rate of rice seeds is closely related to temperature.
    5.2 Water absorption generally increases as temperature increases.
    5.3 The relationship can be expressed as: unsaturated water absorption (%) = 14.289T-10.719 (where T is
temperature)
```

Figure A6: **The second example for few-shot structurization.**

```
Source Statement {
```
The amount of hype and uneducated investors/speculators driving its prices up. Just by that, I would say its prices
are inflated. Bear in mind that Facebook doesn't sell anything tangible. They can go down as fast as they went up.
Most of their income is ad-based and single-product oriented, and as such highly dependent on usage and trends
(remember MySpace?). Having said that, all the other "classic" valuation techniques are still valid and you should
utilize them.
```

}

Qwen_max's Output {
## Statement's scope:
```Valuation concerns for Facebook due to hype and lack of tangible products```

## Statement's main aspects and corresponding descriptions:
```
1. Hype and uneducated investors/speculators
    1.1. The amount of hype and uneducated investors/speculators is significant, which drives up the prices.
    1.2. This suggests that the current prices are inflated based on this factor alone.
2. Lack of tangible products
    2.1. Facebook doesn't sell any tangible goods, making its value potentially unstable.
    2.2. Prices can drop rapidly, similar to their rate of increase.
3. Ad-based income and dependency
    3.1. Most of Facebook's income is ad-based and focused on a single product.
    3.2. Their revenue is highly dependent on usage trends and popularity (citing the decline of MySpace).
4. Classic valuation techniques
    4.1. Despite these concerns, traditional valuation methods remain valid for assessing Facebook's value.
    4.2. It's recommended to utilize classic valuation techniques when evaluating the company.
```

}
```

Figure A7: An example for source inputs and structurized results from few-shot Qwen-max.

```
Qwen_7B's Output {
## Statement's scope:
```market analysis of Facebook stock```

## Statement's main aspects and corresponding descriptions:
```
1. Influence of hype and uneducated investors on stock prices
    1.1 The amount of hype and uneducated investors driving its prices up leads to an inflated valuation of Facebook
stock.
    1.2 The price increase can be unstable, as the company's value is dependent on user growth and trends.
2. Evaluation of Facebook's business model
    2.1 Facebook's revenue is primarily generated through advertising and is highly dependent on user behavior and
trends.
    2.2 Their income is single-product oriented, so they are vulnerable to changes in market conditions.
```

Note: The original statement was quite long and complex. It discusses the influence of hype and uneducated investors
on stock prices, before moving on to an evaluation of Facebook's business model. The scope of the statement includes
the analysis of Facebook stock, but it could be more precise by narrowing it down to just "stock prices." This is why
the first part of the rephrased structure includes the phrase "market analysis of Facebook stock." In the second part,
I identified two main aspects: the influence of hype and uneducated investors on stock prices, and the evaluation of
Facebook's business model. Each aspect is further divided into several descriptive sentences that are faithful to the
original statement.
}

LLaMA2_7B's Output {
Scope:
```The reasons behind the high valuation of Facebook```

Main Aspects and Corresponding Descriptions:
1. The influence of hype and uneducated investors/speculators
    1.1 Just by that I would say its prices are inflated.
    1.2 Facebook's reliance on ad-based and single-product-oriented income exposes it to market trends and usage.
2. Dependence on usage and trends
    2.1 Remember MySpace?
3. Validity of alternative valuation techniques
    3.1 Having said that, all the other "classic" valuation techniques are still valid and you should utilize them.

}
```

Figure A8: Corresponding structurization results from few-shot Qwen-7B and LLaMA2-7B.

```
StruXGPT_7B_Qwen's Output {
## Statement's scope:
```The valuation of Facebook's stock considering unpractical factors```

## Statement's main aspects and corresponding descriptions:
```
1. Price inflation due to hype and inexperienced investors/speculators.
    1.1. The vast amounts of hype surrounding Facebook's stock drive its prices up.
    1.2. This suggests that the materiality of its assets is not justified.
2. Facebook's financial stability and potential for volatility.
    2.1. Facebook doesn't sell any tangible goods.
    2.2. They can decline as rapidly as they have grown.
    2.3. The majority of its income comes from advertising and a single product model, making it highly dependent on
usage trends and fashion (parallels with MySpace's decline).
3. Validity and application of valuation techniques.
    3.1. All traditional valuation methods are still applicable to Facebook's stock.
    3.2. Investors are advised to utilize these techniques despite the platform's unique situation.
```
}

StruXGPT_7B_LLaMA's Output {
## Statement's scope:
```Misconceptions about the market value of Facebook based on uneducated sentiment and issues related to its business
model```

## Statement's main aspects and corresponding descriptions:
```
1. Hype and uneducated investors
    1.1. The amount of hype around Facebook is driving up its price.
    1.2. Uneducated investors/speculators are also contributing to the inflated prices.
2. Income dependence
    2.1. Over 50% of Facebook's income comes from advertisements.
    2.2. A significant portion of Facebook's revenue is reliant upon usage and trends.
3. Business model volatility
    3.1. The company does not sell any tangible products, hence its value is inherently volatile.
    3.2. As such, gains can be just as quick as they were achieved.
4. Classic valuation techniques
    4.1. Despite hype and business unpredictability, other "classic" valuation techniques remain applicable to assess
its market value.
    4.2. They should continue to be utilized for a comprehensive assessment of Facebook's worth.
```
}
```

Figure A9: Corresponding structurization results from our fine-tuned StruXGPT-7B-Qwen and StruXGPT-7B-LLaMA.

