# OpenReview forum: "Enhancing LLM’s Cognition via Structurization"
_NeurIPS.cc/2024/Conference — NeurIPS 2024 poster_

### Official Review · Reviewer_eQho · 2024-07-06

**Soundness:** 3
**Presentation:** 3
**Contribution:** 2
**Rating:** 6
**Confidence:** 5

**Summary:**

The paper presents a method to improve the cognitive capabilities of large language models (LLMs) by organizing input context into a hierarchical structure. The method is called context structurization, and it involves transforming unordered contextual sentences into hierarchically structured elements to mimick human cognitive processes. The paper shows the effectiveness of the approach through evaluations on various NLP tasks, such as context-based question-answering, hallucination evaluation, and passage-level dense retrieval, using different model architectures and sizes. The study shows performance improvements and introduces StruXGPT-7B, a distilled model, to perform structurization efficiently.

**Strengths:**

- The study covers a wide range of tasks, providing a comprehensive assessment of the method's effectiveness.
- I like the idea of distilling the structurization capability into a smaller model (StruXGPT-7B).
- Even though the approch lack a bit of novelty it seems to work.

**Weaknesses:**

- The performance improvements are heavily dependent on the quality of the structurization process, so a poor structurization could lead to suboptimal model performance or even confusion.
- Even though there is a performances improvement the process of structurizing context might introduce additional computational overhead.
- The approach lack a bit of novelty, it doesn't introduce fundamentally new concepts and resembles methodologies from previous studies.
- The study relies on metrics such as ROUGE-L and human evaluation, that are useful but they might not fully capture the more complex improvements, such as a cognitive-like processing that structurization aims to achieve.

**Questions:**

- How does the proposed three-layer structure handle texts with more complex relationships or multiple topics?
- What are the computational costs associated with context structurization?

**Limitations:**

- The study heavily relies on specific models like LLaMA2 and Qwen, so i wonder about the generalizability of the results to other LLMs.
- The structurization may be less effective working with extremely long contexts or when the context cannot be easily divided into clear hierarchical segments.

---

> ### Author Rebuttal · Authors · 2024-08-07
>
> Dear reviewer eQho:
>
> We thank the reviewer for the valuable time and constructive suggestions, and our point-to-point responses are presented below:
>
> > **W1**: The performance improvements heavily depend on the quality of the structurization process. Poor structurization can lead to suboptimal model performance or confusion.
>
> **A**: Thanks for pointing out this.
> As performance variation on specific tasks and input samples is a trivial problem, our aim is to build statistically consistent improvements across tasks, datasets, and models.
> Despite that, we also statistic the performance variance in the Qasper dataset when taking our structurized context as input. As reported below, our method merely causes degradation on 3.5% of samples (with relatively lower structurization quality), but ultimately receives a +3.6 improvement over all test samples.
>
> |Struct|Declined|Overall|
> |:--|:--:|:--:|
> |default|3.5%|+3.6|
> |filtered|**3.0%**|**+3.7**|
>
> In addition, as BERTScore has showcased an efficient but reliable proxy measure of the structurization quality, we can filter out the structurization results with a low BERTScore (e.g., lower than 0.05) and take back the original context as input. In this way, the degradation can be alleviated (from 3.5% to 3.0%), further improving the final enhancement to +3.7.
>
> We will add the experiments and discussions.
>
> > **W2/Q2**: While there is a performance improvement, the process of organizing context may lead to additional computational overhead.
>
> **A**: As discussed in Appendix C, the increased inference cost is a common limitation for methods using LLMs for augmentation., which depends on computing resources, acceleration techniques, and data patterns. In comparison with summary-based competitors, our experiments on the question-answering task in Table A5 show that our method significantly outperforms competitors at a comparable extra cost. To investigate the impact of model size, we use LLaMA2-7B/70B as baseline models for hallucination evaluation, with the results displayed here for convenient reference:
>
> |LLM-Evaluator|Enhancement|Total Cost|Extra Cost (%)|
> |:--|:--:|:--:|:--:|
> |LLaMA2-7B|-|1.4s|-|
> |+Ours|+6.4|3.5s|150%|
> |LLaMA2-70B|-|17.6s|-|
> |+Ours|+4.3|19.7s|12%|
>
> Our method showcases more efficiency (e.g., 12% extra cost in trade of +4.3 improvement) when facilitating larger baseline models.
>
> > **W3**: The approach lack a bit of novelty.
>
> **A**: There have been several concurrent papers utilizing structurization-like techniques to enhance LLMs through either retrieval- or prompt-augmented generation[1-4], and our method has two core improvements over the rivals:
>
> 1. **Our method is proven effective across various NLP tasks**.
> 2. **We presented an affordable and scalable StruXGPT-7B model for structurization**.
>
> Due to the space limit, please kindly refer to our response to Reviewer yxZb (R3)'s Weakness 3 (W3) for more details.
>
> > **W4**: The study relies on metrics such as ROUGE-L and human evaluation, that are useful but they might not fully capture the more complex improvements, such as a cognitive-like processing that structurization aims to achieve.
>
> **A**: If we misunderstand your concern, please feel free to correct us immediately. We assume that you are worrying about how to effectively evaluate the structurization quality of our StruXGPT model besides the ROUGE-L and human evaluation metrics.
> In Table 4, we also provide AppEval (improvements on downstream applications, which is our goal to achieve) and BERTScore (semantical similarity for raw and structurized texts) as proxy measures, because they show a high degree of consistency with human evaluation.
> We believe our systematical evaluation metrics can better capture the improvements of our structurization process.
>
> > **Q1/L2**: How does the proposed three-layer structure handle texts with more complex relationships or multiple topics? The structurization may be less effective working with extremely long contexts.
>
> **A**: Thanks for pointing out this. To avoid information loss for very long contexts (e.g., with 32K length), we automatically split the raw text into several chunks (e.g., using paragraph identifiers like `\n\n`), perform structurization on those chunks in parallel, and integrate the structurized chunks into one structure (by concatenating the aspects in those chunks) to capture the whole context. This is exactly our strategy when handling the context in the MuSiQue subset from the LongBench dataset. As for the text with more relations or multiple topics, we can adopt a similar strategy to construct the semantic structure in a bottom-up manner. And as discussed in our paper, we will continue to explore more flexible approaches (such as a MindMap) to capture complicated structures in our future work.
>
> > **L1**: The study heavily relies on specific models like LLaMA2 and Qwen, so I wonder about the generalizability of the results to other LLMs.
>
> **A**: If we misunderstand your concern, please feel free to correct us immediately. If you worry about the downstream generalizability, we have investigated LLaMA, Qwen, ChatGLM, as well as ChatGPT/GPT4 models across various tasks; as for our structurization model (StruXGPT), we have evaluated different architectures (LLaMA/Qwen) and sizes (1.8B/7B/14B in our new experiments).
> We hope those experiments can help confirm our method's generalizability.
>
> We hope our responses can address the reviewer's concerns, and we are more than happy to provide further explanations if there are additional questions.
>
> Best regards,
>
> Authors
>
> ---
>
> [1] Dong et al. Multi-view Content-aware Indexing for Long Document Retrieval. ArXiv'24.
>
> [2] Sarthi et al. RAPTOR: Recursive Abstractive Processing for Tree-Organized Retrieval. ICLR'24.
>
> [3] Cheng et al. Information Re-Organization Improves Reasoning in Large Language Models. ArXiv'24.
>
> [4] Zhong et al. Achieving >97% on GSM8K: Deeply Understanding the Problems Makes LLMs Perfect Reasoners. ArXiv'24.

---

> > ### Comment · Reviewer_eQho · 2024-08-10
> > **Thank you for your reply.**
> >
> > Your explanations are very clear, and I appreciate that you plan to add the experiments and discussions about W1 in the final version. Because of this, I’ve decided to raise my rating to 6.

---

> > > ### Author Response · Authors · 2024-08-11
> > > **Thanks for your timely feedback**
> > >
> > > Dear Reviewer,
> > >
> > > We sincerely appreciate your acknowledging our responses, and we will continue to polish our paper based on your and other reviewer's constructive suggestions.
> > >
> > > ---
> > >
> > > Best regards,
> > >
> > > Authors

---

### Official Review · Reviewer_yxZb · 2024-07-09

**Soundness:** 3
**Presentation:** 3
**Contribution:** 2
**Rating:** 5
**Confidence:** 4

**Summary:**

This paper proposes a new technique for prompting Large Language Models (LLMs) called StruXGPT. The basic idea is to transform the original prompt into a more structured description of the request which contains three levels of information: Scope, Aspects, and Descriptions. While the **Scope** provides an outline summary of the request, the **Aspects** present an itemized and well-ordered list of topics that are associated with their respective **Descriptions** which provide the details. In the paper, the prompt transformation is achieved by either using another large model or by a smaller model obtained by knowledge distillation. The experiments show that this type of structure can help LLMs find the right information needed to answer the requests, improving performance in question-answering tasks and reducing model hallucinations.

**Strengths:**

The proposed technique is useful and easy to implement. As hallucination is currently one of the main challenges faced by real-world systems that employ LLMs, any technique that reduces it should be considered.

**Weaknesses:**

In the abstract and introduction of the paper, there are claims about how human cognition works that are poorly backed up. This is an area of debate and I believe this discussion could be avoided without any harm to the main point of the paper.

The proposal assumes that the prompt transformation process is an easier task for an LLM than directly addressing the request. Although this seems to be the case for question-answering problems considered, there is no discussion about cases in which this may not be adequate. I believe more experiments with other types of datasets would help us to better understand that.

Finally, as there are other papers investigating similar approaches, the novelty of the approach is not a strong point.

**Questions:**

The legend of Table 1 mentions that the prefix Struct- indicates the data fed into LLMs is structured by StruXGPT-7B. However, I don't see this prefix in the table.

**Limitations:**

Yes, the limitations were well addressed in the paper.

---

> ### Author Rebuttal · Authors · 2024-08-07
>
> Dear reviewer yxZb:
>
> We thank the reviewer for the valuable time and constructive suggestions, and our point-to-point responses are presented below:
>
> > **W1**: In the abstract and introduction of the paper, there are claims about how human cognition works that are poorly backed up. This is an area of debate and I believe this discussion could be avoided without any harm to the main point of the paper.
>
> **A**: Thanks for this suggestion. We will eliminate some debatable descriptions (e.g., "In human cognition, sophisticated text sequences will be processed and consolidated into a structured knowledge tree, with factual elements well-organized hierarchically").
>
> > **W2**: The proposal assumes that the prompt transformation process is an easier task for an LLM than directly addressing the request. Although this seems to be the case for question-answering problems considered, there is no discussion about cases in which this may not be adequate. I believe more experiments with other types of datasets would help us to better understand that.
>
> **A**: Thanks for this suggestion. Here we further evaluate our method on two commonly used multi-choice datasets (i.e., MMLU and BBH).
>
> The MMLU benchmark is a typical scenario, where LLMs are asked to answer questions without context references but requiring world knowledge, such as `In 2016, about how many people in the United States were homeless?`. LLMs have to use their parametric knowledge (learned during large-scale pre-training) to find the answer, and context structurization does not help. If we insist on structurizing the question alone (such as `1. Number of homeless people: in 2016, about how many people in the United States were homeless?`), and feed it into LLM's inputs, the model may be disturbed by the repeated information (such as the `1. Number of homeless people:`) and generate wrong answers.
>
> To quantify the results, we take LLaMA2-7B-Chat as the baseline model and feed the structurized question. The performance variation (measured by OpenCompass protocol) is reported below, which shows our method causes a slight 0.1% decrease on the MMLU benchmark.
>
> |Model|MMLU|BBH|
> |:-------|:---------:|:---------:|
> |LLaMA2-7B-Chat|**45.93**|30.47|
> |+Ours|45.84|**31.30**|
>
> On the other hand, we have also tested another common benchmark, BBH, which is designed to evaluate LLMs' reasoning capability when dealing with several logical sentences/statements. Here is an example of structurized question from BBH:
> ```
> The finishing positions of seven golfers in a tournament:
> 1. Golfers' names: The seven golfers were Ana, Eve, Ada, Dan, Rob, Amy, and Joe.
> 2. Dan's finishing position: Dan finished third in the tournament.
> 3. Ana's finishing position relative to Ada: Ana finished above Ada.
> ...
> 7. Rob's finishing position relative to Joe: Rob finished below Joe.
> Who finished first?
> ```
> In this case, our method can adapt well to highlight the logical relations (by abstractive hints like `Dan's finishing position:`) and boost LLM's reasoning abilities.
> The enhancement on multi-choice datasets is consistent with that on question-answering datasets evaluated in our manuscript, demonstrating our method's generalizability.
>
> In conclusion, we suggest users apply structurization to long-form or logically complex contexts, while taking the original question as inputs when there is no context provided.
>
> > **W3**: Finally, as there are other papers investigating similar approaches, the novelty of the approach is not a strong point.
>
> **A**: There have been recently concurrent papers utilizing structurization-like techniques to reduce LLM hallucinations through two major approaches: retrieval-augmented generation[1][2] and prompt-augmented generation[3][4], and our method presents at least two core improvements over the rivals:
>
> 1. **Our method is proven effective across various NLP tasks**: All of those works are specifically designed for a single task ([1][2] for RAG and [3][4] for reasoning), while our method is verified to consistently enhance various LLMs across the question-answering (including multi-hop reasoning), hallucination evaluation, as well as knowledge retrieval tasks.
> 2. **We presented an affordable and scalable StruXGPT-7B model for structurization**: The concurrent rivals usually adopt LLaMA3-70B/ChatGPT/GPT4 for prompt augmentation, which is neither inference-efficient nor cost-friendly. In contrast, our paper has proposed and validated an efficient and effective solution to train a StruXGPT-7B model for structurization, which is more affordable for deployment and may even exceed the giant teacher LLMs (e.g., with 0% format error in structurization.) We will open-source the model, data, and code soon to further facilitate the research in the LLM community.
>
> We hope the discussion with related works can better establish the contribution and novelty of our paper.
>
> > **Q1**: The legend of Table 1 mentions that the prefix Struct- indicates the data fed into LLMs is structured by StruXGPT-7B. However, I don't see this prefix in the table.
>
> **A**: Thanks for pointing out that. It is a typo and we will fix it in our revised manuscript.
>
> We hope our responses can address the reviewer's concerns, and we are more than happy to provide further explanations if there are additional questions.
>
> Best regards,
>
> Authors
>
> ---
>
> [1] Dong et al. Multi-view Content-aware Indexing for Long Document Retrieval. ArXiv'24.
>
> [2] Sarthi et al. RAPTOR: Recursive Abstractive Processing for Tree-Organized Retrieval. ICLR'24.
>
> [3] Cheng et al. Information Re-Organization Improves Reasoning in Large Language Models. ArXiv'24.
>
> [4] Zhong et al. Achieving >97% on GSM8K: Deeply Understanding the Problems Makes LLMs Perfect Reasoners. ArXiv'24.

---

### Official Review · Reviewer_8sM9 · 2024-07-10

**Soundness:** 3
**Presentation:** 4
**Contribution:** 3
**Rating:** 8
**Confidence:** 3

**Summary:**

The paper presents a novel approach to improve the cognitive capabilities of large language models (LLMs) without inferring the model by structuring contextual information hierarchically. The authors propose transforming plain, sequential text into a structured format, enabling LLMs to process and understand complex information more effectively. This method is empirically tested across various NLP tasks, demonstrating consistent performance gains. Additionally, the paper introduces StruXGPT, a distilled 7B model that efficiently executes the structurization process, enhancing the results on context-based QA benchmark, Hallucination Evaluation and Passage-level dense retrieval.

**Strengths:**

1. While structurization of text is not a novel concept itself, the development of StruXGPT, a smaller and more efficient model for structurization, is a significant contribution and it demonstrates the practical applicability of the proposed method.
2. Evaluation and experiments are strong parts of paper, authors provide many detailed experiments with proper comparisons. The demonstrate stable performance gains using proposed method.
3. The paper thoroughly describes the structurization process, including the use of linguistic markers and transformation templates, ensuring clarity and reproducibility.
4. Authors made the research deeper by analyzing attention mechanism on structured texts.

**Weaknesses:**

1. Lack of parameter-efficiency analysis: authors provide 7B model for structurization process, but it is not clear whether smaller or bigger models would significantly hurt or benefit the approach.
2. While the proposed approach shows stable improvements, it is still a lot of additional parameters, which is a concern for scalability to large amount of texts.

**Questions:**

1. Did you experiment with smaller/bigger models for structurization? If yes, what were the results?
2. Are there specific tasks in which such structurization could be worse?
3. Did I understand correctly that the data for training StruXGPT was gather by quering bigger model? Did you analyze the training set in details and do you think the proposed approach would benefit from training data gather or annotated by humans? I am personally worried here about the hallucinations that could be inherited from the bigger model.
4. Do you plan to publish your StruXGPT model?

Also in Table 1 you write "The prefix Struct- indicates..." but there is not such prefix in the table. Did you mean "+StruXGPT (ours)" here?

**Limitations:**

1. Paper investigates only inference stage
2. Introducing StructXPGT is still additional computation cost, however authors provide the analysis of extra cost in appendix

---

> ### Author Rebuttal · Authors · 2024-08-07
>
> Dear reviewer 8sM9:
>
> We thank the reviewer for the valuable time and constructive suggestions, and our point-to-point responses are presented below:
>
> > **W1/Q1**: Lack of parameter-efficiency analysis: authors provide a 7B model for structurization process, but it is not clear whether smaller or bigger models would significantly hurt or benefit the approach.
>
> **A**: Thanks for this suggestion. We have implemented our StruXGPT on Qwen-1.8B/7B/14B respectively. We follow the setting in our ablation section to investigate the mode size with two evaluation protocols: AppEval (an improvement on the Qasper subset with context structurization) and SemEval (semantic similarity with raw and structurized texts in the validation set, captured by BERTScore). Specifically, AppEval evaluates how much the structurization can enhance baseline models' cognition capability, and BERTScore verifies hallucinations during the structurization process. In addition, we also report the error rate when parsing structurization results from the trained StruXGPT model's outputs (denoted as _FormatError_).
>
> |StruXGPT|AppEval|BERTScore|FormatError|
> |:-|:--:|:--:|:--:|
> |Qwen-1.8B|+2.7|0.299|_5.0%_|
> |Qwen-7B|+3.6|0.313|0.0%|
> |Qwen-14B|**+3.8**|**0.323**|0.0%|
>
> Compared with the 7B model chosen in our paper, the smaller 1.8B model, despite its positive enhancement on downstream applications (+2.7), shows slight inferiority in both AppEval (+2.7 v.s. +3.6) and BERTScore (0.299 v.s. 0.313), and presents 5% error rate when parsing structurization results.
> On the other hand, the 14B model brings further improvement to BERTScore, meaning that the structurization content is relatively more faithful to the original text. But the boost on AppEval is insignificant, reaching the upper bound enhancement of structurization.
>
> Therefore, the chosen 7B model is a good trade-off between model capacity (training/inference efficiency) and structurization quality.
> We will add the experiments and discussions in our revised paper.
>
> > **Q2**: Are there specific tasks in which such structurization could be worse?
>
> **A**: As our method focuses on context structurization to enhance LLM's cognition ability, for the tasks where no context is provided, our method may not bring significant enhancement and even get worse.
> The commonly used MMLU benchmark is a typical scenario, where LLMs are asked to answer questions without context references but requiring their parametric knowledge (learned during large-scale pre-training), and context structurization does not help. If we insist on structurizing the question alone and feed it into LLM's inputs, the model may be disturbed by the introduced information from StruXGPT and generate wrong answers. As shown below, our method causes a slight 0.1% decrease (measured by OpenCompass protocol) when taking LLaMA2-7B-Chat as the baseline model.
>
> |Model|MMLU|BBH|
> |:--:|:--:|:--:|
> |LLaMA2-7B-Chat|**45.93**|30.47|
> |+Ours|45.84|**31.30**|
>
> Besides, we have also tested another common benchmark, BBH, which is designed to evaluate LLMs' reasoning capability when dealing with several logical sentences/statements.
> In this case, our method can adapt well to highlight the logical relations and boost LLM's reasoning abilities by 0.8%.
>
> In conclusion, we suggest users apply structurization to long-form or logically complex contexts, while taking the original question as inputs when there is no context provided.
>
> > **Q3**: Did I understand correctly that the data for training StruXGPT was gather by quering bigger model? Did you analyze the training set in details and do you think the proposed approach would benefit from training data gather or annotated by humans? I am personally worried here about the hallucinations that could be inherited from the bigger model.
>
> **A**: Yes, the training data for StruXGPT is distilled from a bigger commercial model (Qwen-max in our paper), and we agree that our method can benefit from human annotations. However, only a large amount of annotations on structurization data can bring significant benefits, which are usually inefficient and unaffordable. Our current data collection and filtering strategy is sufficient to construct a qualified training dataset.
>
> Here we delve into the quality of our training data by taking BERTScore as a proxy metric, as it shows a high degree of consistency with human annotation (see Table 4 in our manuscript). According to the statistics below, over 94% of the data pairs have a positive BERTScore (normalized by the baseline score of 0.83, and a positive BERTScore presents a benign similarity), demonstrating the high quality of our training data.
>
> |Score|>0.0|>0.1|>0.2|>0.3|
> |:--|:--:|:--:|:--:|:--:|
> |Ratio|94.45%|75.89%|51.43%|32.68%|
>
> Furthermore, we eliminated around 5% of data with negative scores and trained another StruXGPT model, and the evaluation below indicates this part of data does not hurt the final performance.
>
> |Training Data|AppEval|BERTScore|FormatError|
> |:--|:--:|:--:|:--:|
> |vanilla|+3.6|0.313|0.0%|
> |filtered|+3.4|0.316|0.0%|
>
> On the other hand, according to the ablation in Table A1, data quantity may play a vital role in further improving our StruXGPT.
> We can curate more raw texts and prompt teacher LLMs to generate structurization candidates, and apply our filtering strategies to construct a large-scale and high-quality dataset for training.
> We will add the experiments and discussions.
>
> > **Q4**: Do you plan to publish your StruXGPT model?
>
> **A**: Sure! The model, code, and data will be made all public soon, so as to facilitate future research in the LLM community.
>
> > **Q5**: In Table 1 there is not such prefix of _Struct-_. Did you mean "+StruXGPT (ours)" here?
>
> **A**: Yes, it is a typo, and we will fix it in the revised version. Thanks for pointing out that.
>
> We hope our responses can address the reviewer's concerns, and we are more than happy to provide further explanations if there are additional questions.
>
> Best regards,
>
> Authors

---

> > ### Comment · Reviewer_8sM9 · 2024-08-08
> >
> > Thank you for your quick and very detailed reply, full of convincing experiments. I believe the score is already high enough, but you have fully answered all the questions. I find it very interesting that increasing the model size affects results in such a way, implying that probably bigger models learns structurization by itself.
> >
> > Best wishes to your paper, and let me know if I can be of any help.

---

> > > ### Author Response · Authors · 2024-08-09
> > >
> > > Dear Reviewer 8sM9,
> > >
> > > We are glad that our detailed responses and additional experiments have helped to address your concerns and clarify any lingering questions.
> > >
> > > Once again, thank you for your time and valuable feedback.
> > >
> > > Best regards,
> > > Authors

---

### Official Review · Reviewer_7Sjn · 2024-07-21

**Soundness:** 3
**Presentation:** 3
**Contribution:** 2
**Rating:** 5
**Confidence:** 4

**Summary:**

This paper introduces the concept of context structurization to enhance the comprehension capabilities of large language models (LLMs) for long texts. The authors propose summarizing the input text into a three-layer structure of Scope-Aspect-Description using LLMs, and then inputting this three-layer structure as an enhanced version into the LLM. Additionally, the authors propose to optimize a 7B model by distilling the structuring capabilities of giant commercial LLMs to reduce computational costs. The effectiveness of the method is validated through experiments on multiple NLP tasks. The experiments also demonstrate that the fine-tuned 7B model can inherit most of the structuring capabilities of the giant commercial LLMs.

**Strengths:**

The paper is well-written and well-structured, making it easy to understand.

The proposed method is technically sound.

The effectiveness of context structurization has been validated across multiple datasets in Context-based Question-Answering, Context-based Summarization, and Passage-level Dense Retrieval, showing improvements on a number datasets.

**Weaknesses:**

The experiments are not sufficiently comprehensive. Firstly, many tables only compare the scenarios with and without StruXGPT (except for Table A2 and A5), without comparing against other advanced prompt engineering methods. Therefore, it is hard to determine whether StruXGPT enhances the cognitive abilities of LLMs more effectively than other methods. Secondly, the authors did not validate the effectiveness of StruXGPT on state-of-the-art LLMs such as GPT-4, so it is unclear whether GPT-4 would still perform well without context structurization.

There are also some related summarization-based augmentation methods that could be discussed or compared [1].

[1] Cheng, Daixuan, Shaohan Huang, and Furu Wei. "Adapting Large Language Models via Reading Comprehension." In The Twelfth International Conference on Learning Representations.

**Questions:**

How does the number of in-context examples affect the results of structurization, and why was the number set to 2?

**Limitations:**

Yes.

---

> ### Author Rebuttal · Authors · 2024-08-07
>
> Dear reviewer 7Sjn:
>
> We thank the reviewer for the valuable time and constructive suggestions, and our point-to-point responses are presented below:
>
> > **W1**: Many tables only compare the scenarios with and without StruXGPT (except for Table A2 and A5), without comparing against other advanced prompt engineering methods. There are also some related augmentation methods that could be discussed or compared [1].
>
> **A**: In our manuscript, we mainly focus on consistent improvements across downstream tasks with our StruXGPT, and also compared with summary-based strategies for question-answering (as mentioned in Table A2 and A5), as well as the popular chain-of-thought (CoT) technique for hallucination evaluation (in Table 2). Below is a simple copy of the experimental results built on top of GPT-3.5-Turbo on the AttrScore dataset:
>
> |Evaluator|Attr.|Contra.|Extra.|Average|
> |:--|:--:|:--:|:--:|:--:|
> |GPT-3.5-Turbo|72.0|30.4|71.7|58.0|
> |GPT-3.5-Turbo + Ours|**77.1**|**31.8**|**77.4**|**62.1**|
> |GPT-3.5-Turbo + CoT|76.4|35.3|74.4|62.0|
> |GPT-3.5-Turbo + CoT + Ours|**78.9**|**42.9**|**74.5**|**65.4**|
>
> Accordingly, our method presents comparable performance against CoT, and more importantly, our method does not conflict with existing advanced prompt engineering methods.
> As shown in the above table, we achieves better enhancement after integrating with CoT, illustrating the compatibility and extensibility to more advanced strategies.
>
> Here, we further compare with AdaptLLM[1] on the BoolQ dataset for reading comprehension. In particular, AdaptLLM developed several domain-specific LLM (in BioMedicine, Finance, and Low) via the proposed training technique, which however causes inferiority in general reading comprehension capability. Compared to the baseline model (LLaMA-7B), AdaptLLM-Fin does not introduce significant boosts, while AdaptLLM-Bio/Law even cause performance drops, which is mainly because AdaptLLM's domain-adaptation tuning will harm the general capability more or less.
> In contrast, our method does not alter the baseline model, but only structurizes the input context to enhance LLM's cognition ability on downstream tasks, showing stable and consistent improvements (e.g., a 2.5% increase on BoolQ).
>
> |Dataset|Metric|Baseline|AdaptLLM (Bio/Fin/Law)|Ours|
> |:--|:--:|:--:|:--:|:--:|
> |BoolQ|Acc|55.7|50.7 / 55.8 / 53.9|**58.2**|
>
> We will add those comparisons and discussions in our revised manuscript.
>
> > **W2**: The authors did not validate the effectiveness of StruXGPT on state-of-the-art LLMs such as GPT-4.
>
> **A**: Thanks for this suggestion. As discussed above, we have evaluated our efficacy on the powerful GPT-3.5-Turbo model in our manuscript, and here we further extend StruXGPT to GPT-4-Turbo to investigate the effectiveness:
>
> |Evaluator|Attr.|Contra.|Extra.|Average|
> |:--|:--:|:--:|:--:|:--:|
> |GPT-4-Turbo|86.2|43.3|88.3|72.6|
> |GPT-4-Turbo + Ours|**87.6**|**48.3**|**89.7**|**75.2**|
> |GPT-4-Turbo + CoT|**88.8**|48.9|89.7|75.8|
> |GPT-4-Turbo + CoT + Ours|88.5|**52.8**|**90.3**|**77.2**|
>
> As shown above, our method presents consistent benefits for powerful GPT-3.5 and GPT-4 models, with or without the CoT strategy.
> We believe the experiments and discussions can further validate the effectiveness of our StruXGPT approach.
>
> > **Q1**: How does the number of in-context examples affect the results of structurization, and why was the number set to 2?
>
> **A**: In our paper, we choose 2 in-context examples to prompt commercial LLMs (as a teacher) to generate data pairs of raw/structurized text for training our StruXGPT-7B model (as a student). We think it is enough for teacher models to understand the structurization process and generate valid training samples, as the 2 examples respectively describe the 2 most common types of real-world text (i.e., with/without existing indicators like `1`, `2`, etc).
>
> To further verify it, we investigate the number of in-context examples for teacher models with two evaluation protocols (as in Table 4 in our manuscript): AppEval (an improvement on Qasper subset with context structurization) and SemEval (semantic similarity with raw and structurized texts in the validation set, captured by BERTScore). Specifically, AppEval evaluates how much the structurization can enhance baseline models' cognition capability, and BERTScore verifies hallucinations during the structurization process. Besides, we also report the error rate when parsing structurization results from the teacher model's outputs (denoted as _FormatError_). We respectively adopt 1/2/3 few-shot examples to evaluate the structurization quality, and the results are displayed as follows:
>
> |nShot|AppEval|BERTScore|FormatError|
> |:--|:--:|:--:|:--:|
> |1-shot|+1.8|0.282|25.4%|
> |2-shot|+3.2|**0.308**|7.4%|
> |3-shot|**+3.3**|0.302|**5.5%**|
>
> According to the results, 1-shot is apparently insufficient to illustrate structurization, while 2- and 3-shot achieve comparable structurization quality evaluated by AppEval and BERTScore.
> Notably, 3-shot receives a 2% lower FormatError than 2-shot, in trade for the increased inference cost (because of increased few-shot samples).
> We argue that for the final StruXGPT training, the 2% gap (around 400 samples from 22K in total) does not make a difference, as we will eliminate the samples with the wrong structurization format from the training set.
>
> In conclusion, we recommend users to apply 3- or even more shots when prompting teacher LLMs if available, otherwise 2-shot is also a good choice to balance the inference cost and structurization quality. We will add the experiments and discussions in our revised paper.
>
> We hope our responses can address the reviewer's concerns, and we are more than happy to provide further explanations if there are additional questions.
>
> Best regards,
>
> Authors
>
> ---
>
> [1] Cheng, Daixuan, Shaohan Huang, and Furu Wei. "Adapting Large Language Models via Reading Comprehension." In The Twelfth International Conference on Learning Representations.

---

> > ### Comment · Reviewer_7Sjn · 2024-08-12
> > **Response for the rebuttal**
> >
> > I appreciate the authors' responses to my questions, which addressed part of my concerns. However, merely comparing chain-of-thought (CoT) and only comparing AdaptLLM on a single dataset is not sufficiently convincing for me. In AdaptLLM paper, there are other datasets where AdaptLLM performs significantly better than the baseline. How does the authors' method perform on these datasets?

---

> > > ### Author Response · Authors · 2024-08-13
> > >
> > > Dear Reviewer 7Sjn,
> > >
> > > We sincerely thank you for your feedback.  As it takes us some time to re-produce AdaptLLM's results and implement our method, we have currently compared with AdaptLLM on three subsets respectively in the medicine/finance/law domain.
> > >
> > > |  Domain  |   Subset  |    Metric   |  Baseline |  AdaptLLM |    Ours   |
> > > |:--------|:---------|:-----------|:---------|:---------|:---------|
> > > | Medicine |  PubMedQA |     Acc     |    59.6   |    **63.3**   |    63.0   |
> > > |  Finace  | ConvFinQA |      EM     |    29.2   |    **41.5**   |    36.5   |
> > > |    Law   |   SCOTUS  | mic-F1/mac-F1 | 28.3/10.8 | 30.0/**17.8** | **30.6**/15.6 |
> > >
> > > According to the results above, our method can also boost the Llama-7b baseline for 3%-7% **without training**, while AdaptLLM requires _costly continual training_ of the baseline model on each domain corpus. Although our final performance is slightly inferior to the domain-specialized AdaptLLM, our **generalizability** emphasizes the contribution of our work (we bring _consistent enhancement across downstream domains_ and cause _no degradation on general tasks_, as stated in our previous response).
> > >
> > > Due to the time limitation, we can temporally present the performance on three subsets (plus the aforementioned extra general subset).  In our revised paper, we will provide further comparisons on other datasets and with other approaches to emphasize our method's efficacy.
> > >
> > > We hope those results can address your remaining concerns.  And if there is any further question, please do not hesitate to tell us.
> > >
> > > ---
> > >
> > > Best regards,
> > >
> > > Authors

---

> > > > ### Comment · Reviewer_7Sjn · 2024-08-13
> > > > **Response for the rebuttal**
> > > >
> > > > Thank you for providing more results. I have raised my score. I look forward to seeing more comparisons with other approaches.

---

> > > > > ### Author Response · Authors · 2024-08-13
> > > > >
> > > > > Dear Reviewer 7Sjn,
> > > > >
> > > > > Thank you for your positive feedback and for raising your score! We will include more comparisons in the final version.
> > > > >
> > > > > ---
> > > > >
> > > > > Best regards,
> > > > >
> > > > > Authors

---

### Author Rebuttal · Authors · 2024-08-07

We thank all reviewers for their valuable time and constructive suggestions when evaluating our manuscript. We are really encouraged to see **ALL** reviewers find our method **technically solid**, **extensively validated**, and **well-presented**.

We have provided point-to-point responses to reviewers' comments below, and here is a brief summary of the included experiments and explanations:

* **Comparisons with other augmentation approaches**. We have presented further comparisons with the recent AdaptLLM approach, and extended to incorporating the GPT-4 model as well as the CoT technique to demonstrate our method's efficacy and compatibility.

* **Ablations on training corpus and model size for StruXGPT**. We have additionally investigated the impact of StruXGPT's model size on structurization quality, and also quantify and guarantee the data quality for StruXGPT's training corpus with in-depth analysis.

* **Discussion with concurrent structurization works**. We have discussed several concurrent works to emphasize our novelty and contribution: effectiveness across various downstream models and tasks with a unified structurization, and the affordable and scalable StruXGPT model for this structurization.

* **Exploration of structurization's capability boundary**. We have supplemented extra evaluations on common benchmarks (such as MMLU and BBH) to further study the capability boundary of our method, so as to provide practical suggestions in real-world applications.

We believe reviewers' comments have made our paper much stronger, and we hope our work can further inspire the LLM community to a deeper study in model cognition and generalized artificial intelligence via structurization.

---

### Decision · Program_Chairs · 2024-09-25

**Decision:**

Accept (poster)

**Comment:**

This paper introduces enhancing the comprehension capabilities of LLMs by structuring contextual information hierarchically. The authors propose a three-layer structure to summarize input text into a Scope-Aspect-Description format. They optimize a 7B model (StruXGPT) by distilling the structuring capabilities of LLMs to reduce computational costs. The method improves performance in some NLP tasks, including QA and passage-level dense retrieval, and the fine-tuned 7B model inherits most of the structuring capabilities of large LLMs. The aurhors findings also suggest that the structured prompt transformation reduces model hallucinations and improves performance in context-based QA benchmarks.
There were several questions about the experiments, and strengthening of the findings through better ablations was asked. The authors have provided additional evidence by including different model sizes, and included additional evidence on common benchmarks (such as MMLU and BBH). Reviewers have asked comparison to different models, which authors have agreed to include in the final submission.